# LICORICE: LABEL-EFFICIENT CONCEPT-BASED INTERPRETABLE REINFORCEMENT LEARNING

**Zhuorui Ye**[*][†]
Institute for Interdisciplinary Information Sciences
Tsinghua University
Beijing, China
cuizhuyefei@gmail.com

**Stephanie Milani**[*]
Machine Learning Department
Carnegie Mellon University
Pittsburgh, PA 15213
smilani@cs.cmu.edu

**Geoffrey J. Gordon**
Machine Learning Department
Carnegie Mellon University
Pittsburgh, PA 15213
ggordon@cs.cmu.edu

**Fei Fang**
Software and Societal Systems Department
Carnegie Mellon University
Pittsburgh, PA 15213
feifang@cmu.edu

## ABSTRACT

Recent advances in reinforcement learning (RL) have predominantly leveraged neural network policies for decision-making, yet these models often lack interpretability, posing challenges for stakeholder comprehension and trust. Concept bottleneck models offer an interpretable alternative by integrating human-understandable concepts into policies. However, prior work assumes that concept annotations are readily available during training. For RL, this requirement poses a significant limitation: it necessitates continuous real-time concept annotation, which either places an impractical burden on human annotators or incurs substantial costs in API queries and inference time when employing automated labeling methods. To overcome this limitation, we introduce a novel training scheme that enables RL agents to efficiently learn a concept-based policy by only querying annotators to label a small set of data. Our algorithm, LICORICE, involves three main contributions: interleaving concept learning and RL training, using an ensemble to actively select informative data points for labeling, and decorrelating the concept data. We show how LICORICE reduces human labeling efforts to 500 or fewer concept labels in three environments, and 5000 or fewer in two more complex environments, all at no cost to performance. We also explore the use of VLMs as automated concept annotators, finding them effective in some cases but imperfect in others. Our work significantly reduces the annotation burden for interpretable RL, making it more practical for real-world applications that necessitate transparency. Our code is released.[1]

## 1 INTRODUCTION

In reinforcement learning (RL), agents learn a *policy*, which is a strategy for making sequential decisions in complex environments. Agents typically represent the policy as a neural network, as this representation tends to yield high performance (Mirhoseini et al., 2021). However, this choice can come at a cost: such policies are challenging for stakeholders to interpret — particularly when the network inputs are also complex, such as high-dimensional sensor data. This opacity can pose a significant hurdle, especially in applications where understanding the rationale behind decisions is critical, such as healthcare (Yu et al., 2021) or finance (Liu et al., 2022). In such applications, decisions can have significant consequences, so it is essential for stakeholders to fully grasp the reasoning behind actions to confidently adopt or collaborate on a policy.

---

[*]Equal Contribution.
[†]This work was done when Ye was a visiting intern at CMU.
[1]https://github.com/cuizhuyefei/LICORICE

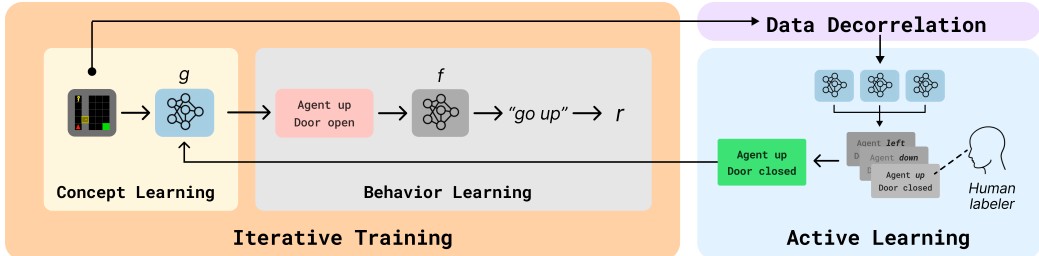

Figure 1: **LICORICE overview.** In concept-based RL, the policy first maps from states to the concepts in a bottleneck layer with $g$, and then maps from concepts to (distributions over) actions with $f$. During training, LICORICE addresses concept label efficiency concerns with three key components: i) iterative training, ii) data decorrelation, and iii) active learning.

To address interpretability concerns in supervised learning, recent works have integrated human-understandable concepts into neural networks through concept bottleneck models (Koh et al., 2020; Espinosa Zarlenga et al., 2022). These models insert a bottleneck layer whose units correspond to interpretable concepts, ensuring that the final decisions depend on these concepts instead of on opaque raw inputs. By training the model both to have high task accuracy and to accurately match experts' concept labels, these models learn a high-level concept-based representation that is simultaneously meaningful to humans and useful for machine learning tasks. As an example, a concept-based explanation for a bird classification task might include a unit that encodes the bird's wing color. Concepts also offer a path to compositional generalization by enabling subtask decomposition into meaningful, reusable components (Mao et al., 2022; Wang et al., 2023).

More recently, these techniques have been applied to RL by incorporating a concept bottleneck in the policy (Grupen et al., 2022; Zabounidis et al., 2023), so that the chosen actions are a function of the human-understandable concepts. However, a significant challenge emerges in this method's practical application: past work assumes that concept annotations are readily available during RL training. To learn a mapping from states to concepts, an RL agent requires concept information for every state encountered during training, which is often millions or billions of state-action pairs. Because many domains do not automatically provide these concept labels (e.g., autonomous driving), we require a means of obtaining them. Relying on human labelers is impractical, risking errors due to fatigue (Marshall & Shipman, 2013). Using vision language models (VLMs) for automated concept extraction (Oikarinen et al., 2023), while alleviating the human bottleneck, introduces significant API or inference costs that can be prohibitive.

In this work, we address this challenge with LICORICE (**L**abel-efficient **I**nterpretable **CO**ncept-based **R**e**I**nfor**CE**ment learning), a novel training paradigm designed to minimize the number of concept annotation queries while maintaining high task performance. Figure 1 illustrates our algorithm. LICORICE tackles three key challenges. First, it addresses the problem of concept learning on off-policy or outdated data. If concepts are learned from data collected by a random policy, the concept distribution may not reflect the data from an optimal policy. To ensure that concept learning occurs on more recent and on-policy data, LICORICE interleaves concept learning and RL training through *iterative training*: it alternately freezes the network layers corresponding to either the concept learning part or the decision-making part. Second, LICORICE addresses the problem of limited training data diversity that occurs when an agent interacts with the environment, thereby generating sequences of highly similar, temporally correlated data points. To tackle this issue, LICORICE implements a *data decorrelation* strategy to produce a more diverse set of training samples. Third, LICORICE addresses the inefficient use of annotation effort, where labeled samples may provide redundant information. To resolve this problem, we employ *disagreement-based active learning* using a concept ensemble to select the most informative data points for labeling.

To evaluate the effectiveness of LICORICE, we conduct experiments in two scenarios—perfect human annotation and VLM annotation—on five environments with image input: an image-based version of CartPole, two Minigrid environments, and two Atari environments. First, with assumption of perfect human annotation, we show that LICORICE yields both higher concept accuracy and higher reward while requiring fewer annotation queries compared to baseline methods. Second, we find that VLMs can indeed serve as concept annotators for some, but not all, of the environments.

Our contributions are summarized as follows:

- To the best of our knowledge, we are the first to investigate the problem of a limited concept annotation budget for interpretable RL. We introduce LICORICE, a novel training scheme that enables label efficient learning of concept-based RL policies.
- We conduct extensive experiments to show the effectiveness of LICORICE across five environments with varying budget constraints.
- We study the use of VLMs as automated concept annotators, demonstrating their effectiveness in certain environments while highlighting challenges in others.

## 2 PRELIMINARIES

**Reinforcement Learning.** In RL, an agent learns to make decisions by interacting with an environment (Sutton & Barto, 2018). The environment is commonly modeled as a Markov decision process (Puterman, 2014), consisting of the following components: a set of states $\mathcal{S}$, a set of actions $\mathcal{A}$, a state transition function $T : \mathcal{S} \times \mathcal{A} \times \mathcal{S} \to [0, 1]$ that indicates the probability of transitioning from one state to another given an action, a reward function $R : \mathcal{S} \times \mathcal{A} \times \mathcal{S} \to \mathbb{R}$ that assigns a reward for each state-action-state transition, and a discount factor $\gamma \in [0, 1]$ that determines the present value of future rewards. The agent learns a policy $\pi : \mathcal{S} \times \mathcal{A} \to [0, 1]$, which maps states to distributions over actions with the aim of maximizing the expected cumulative discounted reward. We evaluate a policy via its value function, which is defined as $V^\pi(s) = \mathbb{E}_\pi[\sum_{k=0}^\infty \gamma^k r_{t+k+1} \mid s_t = s]$. The ultimate aim in RL is to determine the optimal policy, $\pi^*$, through iterative refinement based on environmental feedback.

**Concept Policy Models.** Concept-based explanations have emerged as a common paradigm in explainable AI (Poeta et al., 2023). They explain a model's decision-making process through human-understandable attributes and abstractions. In supervised learning, concept bottleneck models (Koh et al., 2020) implement this approach using two key functions: a concept predictor $g : X \to C$, mapping inputs to interpretable concepts, and an output predictor $f : C \to Y$, mapping the concept predictions to a downstream task space, such as labels for classification. As a result, the prediction takes the form $\hat{y} = f(g(x))$, so that the input $x$ influences the output solely through the bottleneck layer $\hat{c} = g(x)$. In RL, since we want the policy to be interpretable, we insert a concept bottleneck layer into the policy network:

$$\pi(s) = f(g(s)),$$

such that $\pi$ maps from states $s \in \mathcal{S}$ to concepts $c \in C$ to actions $a \in \mathcal{A}$ (Grupen et al., 2022; Zabounidis et al., 2023). This setup allows the policy to base its decisions on understandable and meaningful features. As a result, we can use any RL algorithm that can be modified to include an additional loss function for concept prediction.

**Training Concept Policy Models.** Training these models requires a dataset of state-concept-action triplets $(s, c, a) \in \mathcal{S} \times C \times \mathcal{A}$. The functions $f$ and $g$ are typically implemented as neural networks, with their parameters collectively defined as $\theta$. Previous work (Zabounidis et al., 2023) simply combines the concept prediction loss $L^C(\theta)$ and RL loss $L^{\mathrm{RL}}(\theta)$:

$$L(\theta) = L^{\mathrm{RL}}(\theta) + \lambda_C L^C(\theta),$$

where the exact definitions of $L^{\mathrm{RL}}(\theta)$ and $L^C(\theta)$ depend on the choice of RL algorithm and concept learning task. The objective is to find the optimal parameters $\theta^*$ that minimize this combined loss function with the coefficient $\lambda_C$. However, this approach requires continuous access to ground-truth concepts for training $f$, which may not always be feasible or desirable in practical RL scenarios.

## 3 LICORICE

As we have mentioned, the standard way of training concept-based RL assumes continuous access to an oracle to provide concept labels. However, this assumption is problematic due to the large annotation cost incurred by human or automated labeling efforts. To reduce the number of concept labels required for concept-based RL, we propose LICORICE, a novel algorithm for interpretable RL consisting of three main components: iterative training, data decorrelation, and disagreement-based active learning. The full pseudocode is in Algorithm 1.

---

**Algorithm 1** LICORICE (**L**abel-efficient **I**nterpretable **CO**ncept-based **ReInforCE**ment learning)

---

1: **Input:** Total budget $B$, number of iterations $M$, sample acceptance threshold $p$, ratio $\tau$ for active learning, batch size for querying $b$, number of concept models $N$ to ensemble
2: **Initialize:** training set $\mathcal{D}_{\text{train}} \leftarrow \emptyset$, and validation set $\mathcal{D}_{\text{val}} \leftarrow \emptyset$
3: **for** $m = 1$ **to** $M$ **do**
4:     Allocate budget for iteration $m$: $B_m \leftarrow \frac{B}{M}$
5:     **while** $|\mathcal{U}_m| < \tau \cdot B_m$ **do**
6:         Execute policy $\pi_m$ to collect unlabeled data $\mathcal{U}_m$ using acceptance rate $p$
7:     **end while**
8:     Initialize iteration-specific datasets for collecting labeled data: $\mathcal{D}'_{\text{train}} \leftarrow \emptyset, \mathcal{D}'_{\text{val}} \leftarrow \emptyset$
9:     **while** $B_m > 0$ **do**
10:        Train $N$ concept models $\{\tilde{g}_i\}_{i=1}^N$ on $\mathcal{D}_{\text{train}} \cup \mathcal{D}'_{\text{train}}$, using $\mathcal{D}_{\text{val}} \cup \mathcal{D}'_{\text{val}}$ for early stopping
11:        Calculate acquisition function value $\alpha(s)$ for all $s \in \mathcal{U}_m \setminus (\mathcal{D}'_{\text{train}} \cup \mathcal{D}'_{\text{val}})$
12:        Choose $b_m = \min(b, B_m)$ unlabeled points from $\mathcal{U}_m$ according to $\arg\max_s \alpha(s)$
13:        Query for concept labels to obtain new dataset $\mathcal{D}_m \leftarrow \{(s, c)\}^{b_m}$
14:        Split $\mathcal{D}_m$ into train and validation splits and add to $\mathcal{D}'_{\text{train}}, \mathcal{D}'_{\text{val}}$
15:        Decrement budget: $B_m \leftarrow B_m - b_m$
16:     **end while**
17:     Aggregate datasets: $\mathcal{D}_{\text{train}} \leftarrow \mathcal{D}_{\text{train}} \cup \mathcal{D}'_{\text{train}}, \mathcal{D}_{\text{val}} \leftarrow \mathcal{D}_{\text{val}} \cup \mathcal{D}'_{\text{val}}$
18:     Continue training the concept network $g$ on $\mathcal{D}_{\text{train}}$, using $\mathcal{D}_{\text{val}}$ for early stopping
19:     Freeze $g$ and continue training $f$ using standard RL training to obtain $\pi_{m+1}$
20: **end for**

---

**Iterative Training.** As the agent's policy improves, the distribution of the visited states and their associated concepts changes. Consider an MDP where states are indexed. With a random (initial) policy, the agent tends to visit small-index states near the initial state, encountering only the concepts relevant to those states. However, as the policy improves, the agent explores higher-index states, leading to a shift in both the state and concept visitation distribution. If we exhaust all queries at the beginning of training, we risk training the model only on the concepts associated with small-index states from the random policy, potentially missing important concepts that emerge as the agent explores. We therefore propose iterative training to enable LICORICE to progressively refine its understanding of concepts as the policy improves. Iterative training consists of two parts: concept learning and behavior learning. Assume that we have access to an annotated concept dataset $\mathcal{D}_{\text{train}}$ generated by our current policy $\pi_m$. Then, *concept learning* focuses on training the concept portion of the network $g$ on this dataset $\mathcal{D}_{\text{train}}$ (line 18). *Behavior learning* involves freezing $g$ and training the behavior portion of the network $f$ with any standard RL algorithm with its associated loss $L^{\text{RL}}$ (line 19). Upon completion of behavior learning, we obtain an updated policy $\pi_{m+1}$, which serves to collect more unlabeled concept data to help train $g$ in the next iteration.

**Data Decorrelation.** We do not obtain ground-truth concept labels for all states when rolling out the current policy $\pi_m$. Instead, rollouts produce a dataset $\mathcal{U}_m$ of unlabeled states as candidates for querying for ground-truth concepts from an oracle. Simply collecting and labeling all encountered states would give us long chains of nearly-identical samples, undermining the diversity we need for effective learning. To resolve this challenge, we introduce data decorrelation with two key components: a budget multiplier $\tau$ that sets $|\mathcal{U}_m| = \tau \cdot B_m$, and a per-state acceptance probability $p$. This random acceptance/rejection mechanism leverages a fundamental property of Markov processes—states become increasingly independent as their temporal distance grows according to the mixing time—allowing us to build more diverse datasets from our policy rollouts.

**Disagreement-Based Active Learning.** Equipped with $\mathcal{U}_m$, we are now prepared to select data points for querying for concept labels. The purpose of this stage is to make use of our limited labeling budget $B_m$ to collect a labeled dataset $\mathcal{D}_m$ for training $g$. We propose to train a concept ensemble—consisting of $N$ independent concept models—from scratch on the dataset of training points $\mathcal{D}_{\text{train}}$ that has been aggregated over all iterations (line 10). We use this ensemble to calculate the disagreement-based acquisition function $\alpha(\cdot)$, which determines whether each candidate in $\mathcal{U}_m$ ought to receive a ground-truth concept label (line 11). This function targets samples where prediction disagreement is highest among ensemble members, as these points often represent areas of uncertainty or decision boundaries where additional labeled data would be most informative. For

$\alpha(\cdot)$, we use two different formulations, depending on the concept learning task. For concept classification, we use a query-by-committee approach (Seung et al., 1992), in which we prioritize points with a high proportion of predicted class labels that are not the modal class prediction (also called the variation ratio (Beluch et al., 2018)):

$$\alpha(s) = 1 - \max_{c \in C} \frac{1}{N} \sum_{i=1}^{N} [\tilde{g}_i(s) = c].$$

Here, $\tilde{g}_i(s)$ is the concept prediction of the $i$-th model on state $s$. For concept regression, we directly use the estimate of variance across the concept models as a measure of disagreement:

$$\alpha(s) = \sigma^2(s) = \frac{1}{N-1} \sum_{i=1}^{N} (\tilde{g}_i(s) - \mu(s))^2,$$

where $\mu(s) = \frac{1}{N} \sum_{i=1}^{N} \tilde{g}_i(s)$ is the mean prediction of the $N$ concept models. After querying for $B_m$ ground-truth concept labels (line 13) using $\alpha(\cdot)$, we are prepared to continue training $g$ during the concept learning stage.

**Implementation Details.** For training $g$ (line 18), we employ two types of loss functions depending on the nature of the concepts: MSE for continuous concepts and cross-entropy loss for categorical concepts. If the problem requires mixed-type concepts, we either discretize continuous attributes or simply use a mixed loss. To train $f$ (line 19), we freeze $g$ and use PPO to continue training $f$ from the previous iterations, resulting in the final policy $\pi_{M+1}$. For complex environments that require many iterations, we observe that RL-related parameters may get stuck in a local region in early iterations, becoming hard to optimize later. To mitigate this issue, in the last iteration, we additionally train a new RL network $f'$ from scratch when fixing $g$ with $\pi_{M+1}$ as the anchoring policy to get the final $\pi'_{M+1}$. The anchoring policy ensures $\pi'_{M+1}$ has a similar distribution to the previous one and the annotated observations are still highly relevant. Specifically, we initialize another policy $\pi_\theta$ with the same frozen $g$ parameters and randomly initialized $f$, then train it with the standard PPO loss function and an additional KL-divergence penalty between the current policy and $\pi_{M+1}$:

$$L(\theta) = L^{\text{PPO}}(\theta) + \beta \cdot \text{KL}[\pi_{M+1}(\cdot \mid s), \pi_\theta(\cdot \mid s)].$$

## 4 EXPERIMENTS

In our experiments, we investigate the following questions:

**RQ 1** Under a limited concept annotation budget, does LICORICE enable both high concept accuracy and high environment reward?

**RQ 2** How effective are VLMs as automated concept annotators when used with LICORICE?

**RQ 3** How label efficient is LICORICE compared with other methods?

**RQ 4** Does LICORICE support test-time concept interventions?

### 4.1 EXPERIMENT SETUP

In each experiment, we run each algorithm 5 times, each with a random seed. All algorithms use PPO (Schulman et al., 2017; Raffin et al., 2021) with a concept bottleneck. More implementation details and hyperparameters are in Appendix A.2.

**Environments.** We investigate these questions across five environments: PixelCartPole (Yang et al., 2021), DoorKey (Chevalier-Boisvert et al., 2023), DynamicObstacles (Chevalier-Boisvert et al., 2023), Boxing (Bellemare et al., 2013), and Pong (Bellemare et al., 2013). Each environment includes a distinct challenge and features a set of interpretable concepts describing key objects and properties. We summarize the concepts in Table 1, with more details in Appendix A.1. These environments are characterized by their dual representation: a complex image-based input and a symbolic concept-based representation. PixelCartPole, DoorKey, and DynamicObstacles are simpler because we can extract noiseless ground-truth concept labels from their source code. In contrast, Boxing, as implemented in OCAtari (Delfosse et al., 2023), uses reverse engineering to extract positions of important objects from the game's RAM state. This extraction process introduces a small

| Environment | # Concepts | Concept Type | Binarized # Concepts |
|---|---|---|---|
| PixelCartPole | 4 | Continuous | N/A |
| DoorKey | 12 | Discrete | 46 |
| DynamicObstacles | 11 | Discrete | 30 |
| Boxing | 8 | Discrete | 1480 |
| Pong | 7 | Discrete | 1848 |

Table 1: Summary of environments, including their associated concepts. Counts for the binarized version of the concepts provided where applicable to illustrate the problem size.

amount of noise. The concepts in Pong are derived from the VIPER paper (Bastani et al., 2018) that also uses reverse engineering.

**Baselines.** To our knowledge, there exist no previous algorithms that seek to minimize the number of concept labels for interpretable RL, so we implement three: Sequential-Q, Disagreement-Q, and Random-Q. In Sequential-Q, the agent spends $B$ queries on the first $B$ states encountered during the initial policy rollout. In Disagreement-Q, the agent also spends its budget at the beginning of training; however, it uses active learning with the same $\alpha(\cdot)$ as LICORICE to strategically choose the training data for annotation. In Random-Q, the agent receives $B$ concept labels at random points using a probability to decide whether to query in each state. As a budget-unconstrained baseline, we implement CPM from previous work in multi-agent RL (Zabounidis et al., 2023) for the single-agent setting. As mentioned in §2, CPM jointly trains $g$ and $f$, assuming unlimited access to concept labels, so it also represents an upper bound on concept accuracy.

**VLM Details.** For our VLM experiments, we use GPT-4o (gpt). During training, we query GPT-4o each time LICORICE requires a concept label. As an example, for the concept related to the position of an object, we prompt GPT-4o with instructions to report the coordinates (x y). More details about our prompts can be found in Appendix A.3.

**Performance Metrics.** We compare our reward to an upper bound calculated by training PPO with ground-truth concept labels (PixelCartPole: 500, DoorKey: 0.97, DynamicObstacles: 0.91, Boxing: 86.05, Pong: 21). In the first four environments, percentages (or ratios) make sense since the minimum reasonable reward is 0.[2] In Pong, a random policy has -21 reward so we first get the difference between rewards and -21 and then calculate the ratio. We additionally report the concept error (MSE for regression; $1 -$ accuracy for classification). In Boxing, following the practice of OCAtari, we consider a concept prediction correct if it is within 5 pixels of the ground truth label. In Pong, we simply use classification error. All reported numbers are calculated using 100 evaluation episodes.

## 4.2 RESULTS

### RQ 1: REWARD AND CONCEPT ACCURACY UNDER A LIMITED ANNOTATION BUDGET

**LICORICE achieves similarly high reward and concept performance to the state-of-the-art budget-unconstrained baseline.** We first seek to understand how LICORICE performs compared to the state-of-the-art in concept-based RL, CPM, which is not constrained by concept label budgets. Surprisingly, LICORICE and CPM achieve nearly 100% of the maximum reward in all environments in this unfair comparison (Figure 2). They also achieve a similar concept error rate, only seeing a clear difference in error for the most complex environment, Pong. We emphasize that CPM is given an unlimited budget; in practice, it uses 1M-15M concept labels for all environments, which is around 2000 to $13000\times$ the budget used by LICORICE. In contrast, LICORICE operates under a strict budget constraint, with 3000 labels for Boxing and 5000 for Pong, and at most 500 concept labels for the simpler environments (PixelCartPole: 500, DoorKey: 300, DynamicObstacles: 300).

**LICORICE generally achieves higher reward and lower concept error than budget-constrained baselines.** We study all budget-constrained algorithms (LICORICE plus those in §4.1)

---

[2]In PixelCartPole and DoorKey, all rewards are nonnegative; in DynamicObstacles, an agent can ensure nonnegative reward by simply staying in place; in Boxing, a random policy can get around 0 reward and all of the trained policies have positive reward.

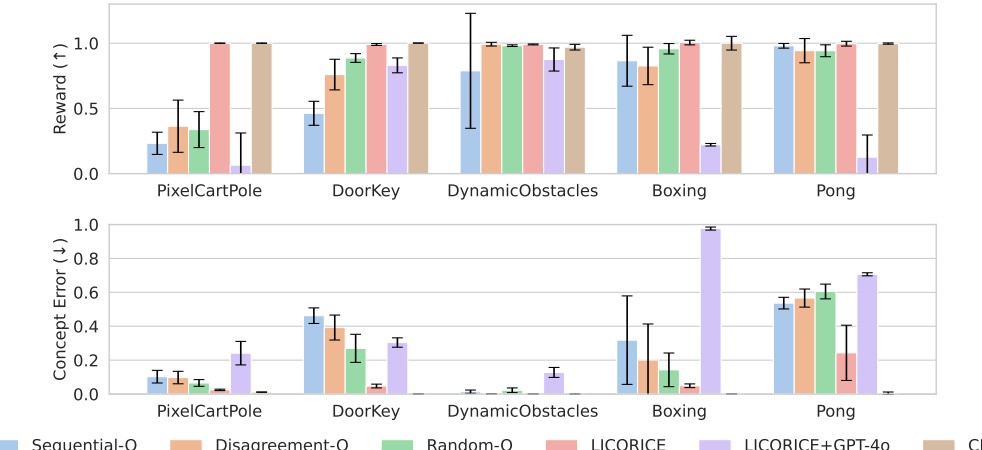

Figure 2: LICORICE generally achieves higher reward and lower concept error than all other budget-constrained algorithms (Sequential-Q, Disagreement-Q, Random-Q). The budgets for each environment are PixelCartpole: 500, DoorKey: 300, DynamicObstacles: 300, Boxing: 3000, Pong: 5000. CPM has an unlimited budget (in practice, it uses 4M, 4M, 1M, 15M, 10M labels, respectively). Despite this extreme budget difference, LICORICE achieves comparable reward and concept error. Full results in Table 5, Appendix B.

under the same fixed concept labeling budget as above. Figure 2 shows that LICORICE outperforms all baselines in terms of reward and concept error on all but arguably the easiest environment, DynamicObstacles. In that environment, LICORICE performs similarly to the baselines. The greatest performance differences occur in PixelCartPole and DoorKey, where LICORICE achieves 100% and 99% of the maximum reward, while the second-best algorithm achieves 36% and 89%, respectively. We therefore answer **RQ 1** affirmatively: not only does LICORICE achieve the same high concept accuracy and high reward as the state-of-the-art in concept-based RL, it also performs on-par to budget-constrained baselines in one environment and outperforms them in the other four.

### RQ 2: VLMs AS CONCEPT ANNOTATORS

We now seek to answer the question of whether VLMs can successfully provide concept labels in lieu of a human annotator within our LICORICE framework. Because using VLMs incurs costs, we operate within the same budget-constrained setting as above. We present these results in Figure 2.

**VLMs can serve as concept annotators for some environments.** In DoorKey and DynamicObstacles, LICORICE with GPT-4o labels achieves 83% and 88% of the maximum reward, respectively. To assess the quality of VLM-generated labels, we compare the concept error rate of our trained model against GPT-4o's labeling error, both evaluated on the same rollout observations, while GPT-4o evaluation is on a random subset of 50 observations, due to API cost constraints. Table 2 shows that the concept error of LICORICE is comparable to that of GPT-4o when GPT-4o labeling error

|  | $c$ Error | |
| Environment | LICORICE+GPT-4o | GPT-4o |
| --- | --- | --- |
| PixelCartPole | 0.25 | 0.24 |
| DoorKey | 0.31 | 0.30 |
| DynamicObstacles | 0.13 | 0.13 |
| Boxing | 0.98 | 0.82 |
| Pong | 0.73 | 0.87 |

Table 2: Concept error comparison. The concept error of LICORICE+GPT-4o matches that of GPT-4o alone in all environments except Boxing.

is not high, which suggests that LICORICE can effectively utilize these concept labels. VLMs, particularly GPT-4o, could serve as concept annotators for certain concepts in certain environments.

**VLMs struggle to provide accurate concepts in more complex environments.** In PixelCartPole, Boxing, and Pong environments, however, GPT-4o struggles to provide accurate labels. In these environments, not only does LICORICE+GPT-4o achieve less than 25% of the maximum reward, it also incurs a large concept error. The challenge with PixelCartPole is the continuous nature of the concepts and the lack of necessary knowledge of physical rules and quantities in this specific

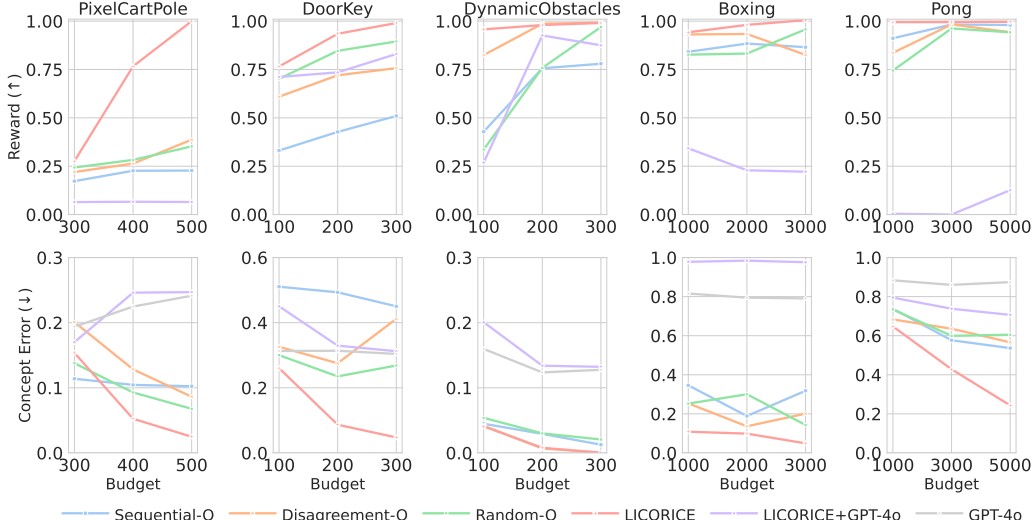

Figure 3: Reward (top) and concept error (bottom) of all algorithms for different budgets. LICORICE more efficiently makes use of the varying budgets, achieving higher reward and lower concept error. Full results with standard deviation are in Tables 6 to 8, Appendix B.

environment. Boxing is particularly challenging due to the large number of possible concept values that each of the 8 discrete concepts can take on (160 or 210 possible values). Similarly, concepts in Pong have hundreds of possible values and some physics quantities require multi-frame inference. We therefore answer **RQ 2** with cautious optimism: there are indeed cases in which LICORICE could be used alongside VLMs, though not all yet. Generally speaking, a less complex environment is more likely to work well with VLMs by enabling clear and detailed labeling instructions in the prompt. However, VLM capabilities need improvement before they can fully alleviate the human annotation burden, especially in safety-critical settings.

RQ 3: INVESTIGATION OF BUDGET REQUIREMENTS FOR PERFORMANCE

Given these results, we now investigate the minimum budget required for all environments to achieve 99% of the reward upper bound. To do so, we incrementally increase the budget for each environment starting from a reasonably small budget until LICORICE reaches 99% of the reward upper bound. In Pong, we further increase the budget until the concept error is below $0.25$. In addition to the baselines, we study LICORICE under perfect (human) annotation and noisy VLM labels in Figure 3.

**LICORICE more rapidly achieves high reward compared to baselines.** Across all environments, LICORICE is the most query-efficient. In three of the environments, it consistently achieves higher reward than baselines across all budget levels. In DynamicObstacles and Pong, LICORICE outperforms baselines at the smallest query budget, after which some baseline method achieves comparable reward. LICORICE is similarly performant with respect to concept error. Except for one budget setting for one environment, LICORICE consistently achieves lower concept error than the baselines.

**For LICORICE+GPT-4o, more concept labels is not always better.** In DoorKey, LICORICE+GPT-4o exhibits predictable behavior in terms of concept error: as the budget increases, the reward increases and the concept error decreases. Counterintuitively, for some environments, the concept error and reward fluctuate with more budget, likely due to the additional labeling noise introduced. We therefore provide a more nuanced answer to **RQ 3**: LICORICE is indeed label efficient across varying budgets, but the benefit is annotator-dependent.

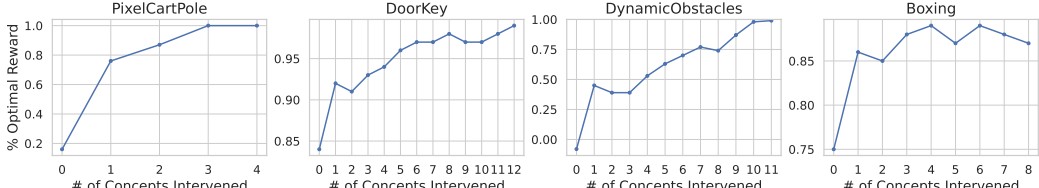

Figure 4: Concept intervention results: LICORICE enables test-time concept intervention.

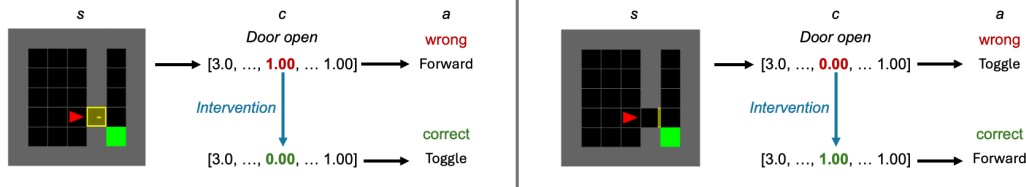

Figure 5: Test-time intervention examples, where intervening on a single concept corrects the action.

RQ 4: INTERPRETABILITY ANALYSIS: TEST-TIME CONCEPT INTERVENTIONS

A great property of concept-based networks is the ability for people to successfully intervene on the concepts to correct them. In RL, this intervention enables the inspection of how different concepts influence the immediate decisions of the agent.

**LICORICE enables test-time concept intervention.** To validate that LICORICE supports test-time concept intervention, we simulate using a noisy concept model $\hat{g}$ with $f$ to study the impact of concept interventions on reward. For PixelCartPole, we add Gaussian noise to each predicted concept with a standard deviation of $0.2$. For the other three environments with discrete concepts, each concept label is randomly changed with probability $0.2$. Following previous work (Koh et al., 2020), we first intervene individually on concepts, setting them to ground truth, and sort the concepts in descending order of reward improvement. We then sequentially intervene on the concepts following this ordering, such that any previously intervened concepts remain intervened. According to Figure 4, using a noisy concept model significantly degrades reward. However, the performance increases with more interventions, meaning LICORICE supports test-time concept intervention, affirmatively answering **RQ 4**.

**Concept interventions show how the policy decisions change.** We now show how domain experts could interact with the model. Specifically, we simulate a counterfactual question: What if a concept value is incorrectly predicted? Figure 5 depicts two examples in DoorKey. Just intervening on the concept of the door being open or closed is sufficient to change the agent's behavior, both highlighting the importance of this specific concept and the ability of a user to intervene on the concepts to interrogate and understand agent behavior.

ADDITIONAL EXPERIMENTS

**Ablation study: all components of LICORICE contribute positively to its performance.** We now conduct an ablation study of our three main contributions: iterative training, decorrelation, and disagreement-based active learning. LICORICE-IT uses only one iteration, LICORICE-DE does not perform data decorrelation, and LICORICE-AC uses the entire unlabeled dataset for querying instead of a subset chosen by active learning. All components are critical to achieving both high reward and low concept error (Table 3; learning curves in Appendix B). However, the component that most contributes to the performance differs depending on the environment. For example, compared with LICORICE, LICORICE-IT exhibits the largest reward gap for PixelCartPole; however, LICORICE-AC yields the largest reward gap for DynamicObstacles, and LICORICE-DE yields the largest gap for DoorKey. This difference may occur because the concepts in DynamicObstacles are simple enough that one iteration is sufficient, so the largest gains can be made with active learning. In contrast, PixelCartPole requires further policy refinement to better estimate on-distribution concept values, so the largest gains can be made by leveraging multiple iterations.

| Algorithm | PixelCartPole | | DoorKey | | DynamicObstacles | | Boxing | |
|---|---|---|---|---|---|---|---|---|
| | $R\uparrow$ | $c$ MSE $\downarrow$ | $R\uparrow$ | $c$ Error $\downarrow$ | $R\uparrow$ | $c$ Error $\downarrow$ | $R\uparrow$ | $c$ Error $\downarrow$ |
| LICORICE-IT | 0.53 | 0.08 | 0.99 | 0.09 | 1.00 | 0.00 | 0.98 | 0.14 |
| LICORICE-DE | 0.97 | 0.03 | 0.78 | 0.20 | 0.98 | 0.00 | 1.00 | 0.04 |
| LICORICE-AC | 0.91 | 0.02 | 0.82 | 0.16 | 0.58 | 0.00 | 0.99 | 0.09 |
| LICORICE | 1.00 | 0.02 | 0.99 | 0.05 | 0.99 | 0.00 | 1.00 | 0.05 |

Table 3: Ablation study results for LICORICE. All components generally help achieve better reward and lower concept error. Full results with standard deviation are in Table 9, Appendix B.

**LICORICE is robust to various hyperparameter values.** LICORICE involves a few key hyperparameters, described in §3 and summarized here for convenience. In data decorrelation, $\tau$ governs the size of the unlabeled dataset collected by the current policy and $p$ controls the rate of acceptance of data points. Active learning involves a concept ensemble with $N$ models. We study the effect of the values of these hyperparameters on DynamicObstacles, finding that LICORICE achieves $96-99\%$ of the maximum reward across 2 values for $N$, 3 values for $p$, and 3 values for $\tau$ (Appendix B.5).

## 5 RELATED WORK

**Interpretable RL.** Interpretable RL has gained significant attention in recent years (Glanois et al., 2024). One prominent area uses rule-based methods—such as decision trees (Silva et al., 2020; Topin et al., 2021), logic (Delfosse et al., 2024), and programs (Verma et al., 2018; Penkov & Ramamoorthy, 2019)—to represent policies. These works either assume that the state is already interpretable or that the policy is pre-specified. Unlike prior work, our method involves *learning* the interpretable representation (through concept training) for policy learning.

**Concept Learning for RL.** Due to successes in the supervised setting (Collins et al., 2023; Sheth & Ebrahimi Kahou, 2023; Zarlenga et al., 2023), concept models have recently been used in RL. Das et al. (2023) trains a joint embedding model between state-action pairs and concept-based explanations to expedite learning via reward shaping. Unlike LICORICE, their policy is not a strict function of the concepts, allowing our techniques to be combined to provide concept-based explanations alongside an interpretable policy. Another example, CPM (Zabounidis et al., 2023), is a multi-agent RL approach that assumes that concept labels are continuously available during training. As we have shown, this approach uses over 1M concept labels, whereas LICORICE achieves similar performance while using $2000\times$ fewer labels.

**Learning Concepts with Human Feedback.** Prior research has explored leveraging human concept labels, but not for RL and without focusing on reducing the labeling burden. For instance, Lage & Doshi-Velez (2020) has users label additional information about the relevance of certain feature dimensions to the concept label to facilitate concept learning for high-dimensional data. In a different vein, Chauhan et al. (2023) develop a test-time intervention policy for querying for concept labels to improve the final prediction. These methods do not directly tackle the issue of reducing the overall labeling burden during training and could be used alongside our method. A concurrent work (Shao et al., 2024) also studies sample efficiency but mainly focus on human task demonstrations.

## 6 DISCUSSION AND CONCLUSION

In this work, we proposed LICORICE, a novel RL algorithm that addresses the critical issue of model interpretability while minimizing reliance on continuous human annotation. We introduced a training scheme that enables RL algorithms to learn concepts efficiently from little labeled concept data. We demonstrated how this approach reduces manual labeling effort. We conducted initial experiments to demonstrate how we can leverage powerful VLMs to infer concepts from raw visual inputs without explicit labels in some environments. There are broader societal impacts of our work that must be considered, including both the impacts of using VLMs in real-world applications and more general considerations around interpretability. For a more detailed discussion of limitations and future work, please refer to Appendix C.

ACKNOWLEDGMENTS

This work was supported in part by NSF grant IIS-2046640 (CAREER). Co-author Fang is supported in part by a Sloan Research Fellowship. We thank NVIDIA for providing computing resources. We thank Zhicheng Zhang for some helpful discussions, Naveen Raman for pointing us to relevant literature, and Rex Chen and Zhiyu (Zoey) Chen for their constructive comments on a draft of this work.

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

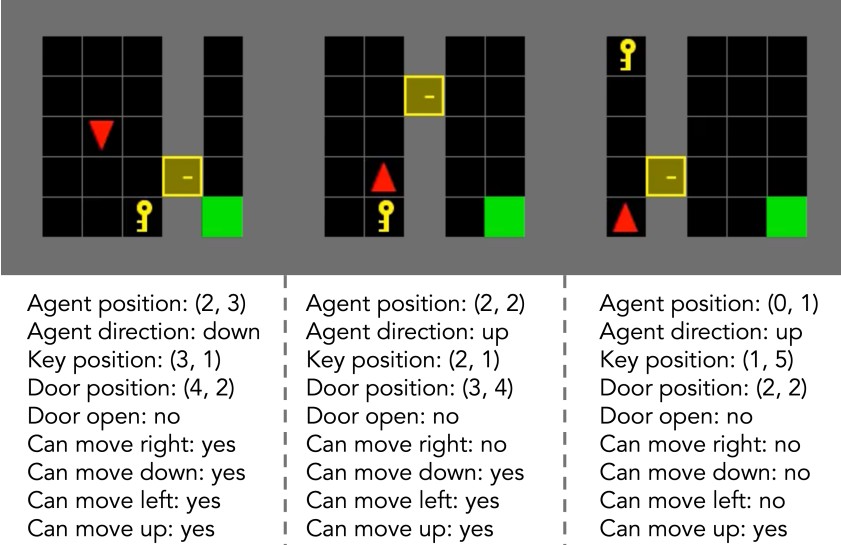

Agent position: (2, 3)  Agent position: (2, 2)  Agent position: (0, 1)
Agent direction: down Agent direction: up  Agent direction: up
Key position: (3, 1)  Key position: (2, 1)  Key position: (1, 5)
Door position: (4, 2)  Door position: (3, 4)  Door position: (2, 2)
Door open: no   Door open: no   Door open: no
Can move right: yes  Can move right: no  Can move right: no
Can move down: yes  Can move down: yes  Can move down: no
Can move left: yes   Can move left: yes   Can move left: no
Can move up: yes   Can move up: yes   Can move up: yes

Figure 6: Example start configurations of the DoorKey environment, along with the associated concepts for each configuration. This shows that concepts must be general enough to apply to many different configurations. As a result, the agent cannot memorize the exact position of the door and key.

## A   EXPERIMENTAL RESULT REPRODUCIBILITY

In this section, we provide descriptions of our concept definitions, prompts, and other experimental details toward the goal of reproducibility.

### A.1   DEFINITIONS OF CONCEPTS

Table 4 provides more details regarding the concepts used in each environment, categorizing them by their names, types, and value ranges. For the PixelCartPole environment, all concepts such as Cart Position, Cart Velocity, Pole Angle, and Pole Angular Velocity are continuous. In contrast, the DoorKey environment features discrete concepts like Agent Position (x and y), Key Position (x and y), and Door Open status, each with specific value ranges. Similarly, the DynamicObstacles environment lists discrete concepts, including Agent and Obstacle positions, with corresponding value ranges.

**Precise concept definitions are crucial for enabling correct agent behavior.** We visualize the start configurations for DoorKey in Figure 6 to illustrate the importance of well-defined concepts in reinforcement learning. DoorKey includes a variety of initial states, each with different positions of the agent, key, and door. This diversity highlights a challenge in concept-based RL: how to define concepts that are both specific enough to be meaningful and general enough to apply across all possible scenarios. An agent trained with poorly defined concepts might perform well in some configurations but fail to generalize to others, leading to suboptimal performance in new or unseen environments. As a result, concept engineering is its own separate but important problem.

### A.2   LICORICE DETAILS.

In this section, we provide additional implementation details for LICORICE. The model architecture has been mostly described in the main text and we state all additional details here. The number of neurons in the concept layer is exactly the number of concepts. For continuous concept values, we directly use a linear layer to map from features to concept values. For discrete concept values, since different concepts have different numbers of categories, we create one linear classification head for each single concept, and to predict the final action, we calculate the class with the largest predicted probability for each concept.

| Environment | Concept Name | Type | Value Ranges | GPT-4o Error |
|---|---|---|---|---|
| PixelCartPole | Cart Position | Continuous | $(-2.4, 2.4)$ | $0.01 \pm 0.00$ |
| | Cart Velocity | Continuous | $\mathbb{R}$ | $0.31 \pm 0.12$ |
| | Pole Angle | Continuous | $(-.2095, .2095)$ | $0.01 \pm 0.00$ |
| | Pole Angular Velocity | Continuous | $\mathbb{R}$ | $0.63 \pm 0.14$ |
| DoorKey | Agent Position x | Discrete | 5 | $0.36 \pm 0.08$ |
| | Agent Position y | Discrete | 5 | $0.41 \pm 0.12$ |
| | Agent Direction | Discrete | 4 | $0.28 \pm 0.06$ |
| | Key Position x | Discrete | 6 | $0.20 \pm 0.03$ |
| | Key Position y | Discrete | 6 | $0.32 \pm 0.11$ |
| | Door Position x | Discrete | 5 | $0.37 \pm 0.07$ |
| | Door Position y | Discrete | 5 | $0.32 \pm 0.04$ |
| | Door Open | Discrete | 2 | $0.38 \pm 0.09$ |
| | Direction Movable Right | Discrete | 2 | $0.30 \pm 0.03$ |
| | Direction Movable Down | Discrete | 2 | $0.19 \pm 0.06$ |
| | Direction Movable Left | Discrete | 2 | $0.33 \pm 0.06$ |
| | Direction Movable Up | Discrete | 2 | $0.20 \pm 0.08$ |
| DynamicObstacles | Agent Position x | Discrete | 3 | $0.03 \pm 0.06$ |
| | Agent Position y | Discrete | 3 | $0.04 \pm 0.06$ |
| | Agent Direction | Discrete | 4 | $0.20 \pm 0.05$ |
| | Obstacle 1 Position x | Discrete | 3 | $0.08 \pm 0.02$ |
| | Obstacle 1 Position y | Discrete | 3 | $0.19 \pm 0.04$ |
| | Obstacle 2 Position x | Discrete | 3 | $0.22 \pm 0.09$ |
| | Obstacle 2 Position y | Discrete | 3 | $0.12 \pm 0.02$ |
| | Direction Movable Right | Discrete | 2 | $0.19 \pm 0.06$ |
| | Direction Movable Down | Discrete | 2 | $0.14 \pm 0.05$ |
| | Direction Movable Left | Discrete | 2 | $0.13 \pm 0.02$ |
| | Direction Movable Up | Discrete | 2 | $0.07 \pm 0.08$ |
| Boxing | Agent Position x at Frame 1 | Discrete | 160 | $0.81 \pm 0.14$ |
| | Agent Position y at Frame 1 | Discrete | 210 | $0.93 \pm 0.05$ |
| | Enemy Position x at Frame 1 | Discrete | 160 | $0.78 \pm 0.13$ |
| | Enemy Position y at Frame 1 | Discrete | 210 | $0.90 \pm 0.07$ |
| | Agent Position x at Frame 2 | Discrete | 160 | $0.67 \pm 0.06$ |
| | Agent Position y at Frame 2 | Discrete | 210 | $0.88 \pm 0.04$ |
| | Enemy Position x at Frame 2 | Discrete | 160 | $0.71 \pm 0.05$ |
| | Enemy Position y at Frame 2 | Discrete | 210 | $0.83 \pm 0.08$ |
| Pong | Ball Position x | Discrete | 176 | $0.99 \pm 0.00$ |
| | Ball Position y | Discrete | 88 | $0.98 \pm 0.00$ |
| | Ball Velocity x | Discrete | 176 | $0.95 \pm 0.00$ |
| | Ball Velocity y | Discrete | 176 | $0.80 \pm 0.00$ |
| | Paddle Velocity | Discrete | 176 | $0.78 \pm 0.05$ |
| | Paddle Acceleration | Discrete | 352 | $0.79 \pm 0.04$ |
| | Paddle Jerk | Discrete | 704 | $0.83 \pm 0.04$ |

Table 4: Concepts and their possible values for all environments. For discrete concepts, we report the number of categories. We also provide the mean GPT-4o labeling error for each single concept, averaged across 5 seeds and corresponding to the same evaluation protocols as the ones with the largest budgets in Table 7. We use MSELoss for continuous values and 1 - accuracy for discrete ones, and in Boxing, we regard one prediction correct if it is within distance 5 to the ground truth. The $\pm$ [value] part shows the standard deviation. The average errors over concepts are $0.24 \pm 0.06$, $0.30 \pm 0.02$, $0.13 \pm 0.03$, $0.82 \pm 0.06$, and $0.88 \pm 0.02$ respectively for the environments.

**Architecture.** The network begins with a CNN-based feature extractor $f_\theta : \mathcal{X} \rightarrow \mathbb{R}^d$, comprising three convolutional layers that map input state $x \in \mathcal{X}$ to a $d$-dimensional feature vector $h = f_\theta(x)$. We then introduce a linear layer $g_\phi : \mathbb{R}^d \rightarrow \mathbb{R}^k$ for concept prediction ($g$), mapping extracted features to $k$ concept values. For regression tasks, each of the $k$ predicted concept is represented as a real value; for classification tasks, each concept maps to one categorical value derived from a classification head. The action prediction component ($f$) is implemented as an MLP with two fully

connected layers, each containing 64 neurons with Tanh activation, taking only $c$ as input to produce action logits $a = \text{MLP}(c)$. Notably, we share the CNN feature extractor $f_\theta$ between policy and value functions, a decision informed by improved performance observed in preliminary experiments.

**Value-Based Methods.** If we were to use a value-based method as the RL backbone, we would need to make the following changes. First, we would need to modify $V(s, a)$ or $Q(s, a)$ to include a concept bottleneck, such that $Q(s, a) = f(g(s))$. Then, we can conduct interleaved training in a similar way to LICORICE.

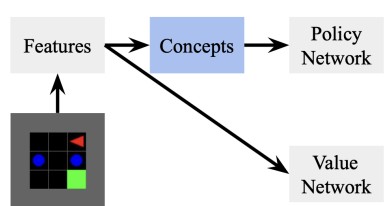

Figure 7: Architecture of our concept-bottleneck actor-critic method.

**Feature Extractor.** If we use the actor-critic paradigm, we propose to share a feature extractor between the policy and value networks, shown in Figure 7. Intuitively, this choice can offer several advantages compared with using image or predicted concepts as input for both networks. Sharing a feature extractor enables both networks to benefit from a common, rich representation of the input data, reducing the number of parameters to be trained. More importantly, it balances the updates of the policy and value networks. In experiments, we observed that directly using the raw image as input for both networks complicated policy learning. Conversely, relying solely on predicted concepts for the value network may limit its accuracy in value estimation, particularly if the concepts do not capture all the nuances relevant to the value predictions.

### A.3 VLM DETAILS

We detail our prompts for each environment here.

**PixelCartPole**

> **Prompt:** Here are the past 4 rendered frames from the CartPole environment. Please use these images to estimate the following values in the latest frame (the last one):
>
> - Cart Position, within the range (-2.4, 2.4)
> - Cart Velocity
> - Pole Angle, within the range (-0.2095, 0.2095)
> - Pole Angular Velocity
>
> Additionally, please note that the last action taken was [last action].
>
> Please carefully determine the following values and give concise answers one by one. Make sure to return an estimated value for each parameter, even if the task may look challenging.
>
> Follow the reporting format:
>
> - Cart Position: estimated_value
> - Cart Velocity: estimated_value
> - Pole Angle: estimated_value
> - Pole Angular Velocity: estimated_value

**DoorKey**

**Prompt:** Here is an image of a 4x4 grid composed of black cells, with each cell either empty or containing an object. Each cell is defined by an integer-valued coordinate system starting at (1, 1) for the top-left cell. The coordinates increase rightward along the x-axis and downward along the y-axis. Within this grid, there is a red isosceles triangle representing the agent, a yellow cell representing the door (which may visually disappear if the door is open), a yellow key icon representing the key (which may disappear), and one green square representing the goal. Carefully analyze the grid and report on the following attributes, focusing only on the black cells as the gray cells are excluded from the active black area.

Detailed Instructions:

1. Agent Position: Identify and report the coordinates (x, y) of the red triangle (agent). Ensure the accuracy by double-checking the agent's exact location within the grid.
2. Agent Direction: Specify the direction the red triangle is facing, which is the orientation of the vertex (pointy corner) of the isosceles triangle. Choose from 'right', 'down', 'left', or 'up'. Clarify that this direction is independent of movement options.
3. Key Position: Provide the coordinates (x, y) where the key is located. If the key is absent, report as (0, 0). Verify visually that the key is present or not before reporting.
4. Door Position:
   - Position: Determine and report the coordinates (x, y) of the door.
   - Status: Assess whether the door is open or closed (closed means the door is visible as a whole yellow cell, while open means the door disappears visually). Report as 'true' for open and 'false' for closed. Double-check the door's appearance to confirm if it is open or closed.
5. Direction Movable: Evaluate and report whether the agent can move one cell in each specified direction, namely, the neighboring cell in that direction is active and empty (not key, closed door, or grey inactive cell):
   - Right (x + 1): Check the cell to the right.
   - Down (y + 1): Check the cell below.
   - Left (x - 1): Check the cell to the left.
   - Up (y - 1): Check the cell above.

   Each direction's feasibility should be reported as 'true' if clear and within the grid, and 'false' otherwise.

Reporting Format: Carefully report each piece of information sequentially, following the format 'name: answer'. Ensure each response is precise and reflects careful verification of the grid details as viewed.

**DynamicObstacles**

**Prompt:** Here is an image of a 3x3 grid composed of black cells, with each cell either empty or containing an object. Each cell is defined by an integer-valued coordinate system starting at (1, 1) for the top-left cell. The coordinates increase rightward along the x-axis and downward along the y-axis. Within this grid, there is a red isosceles triangle representing the agent, two blue balls representing obstacles, and one green square representing the goal. Please carefully determine the following values and give concise answers one by one:

1. Agent Position: Identify and report the coordinates (x, y) of the red triangle (agent). Ensure the accuracy by double-checking the agent's exact location within the grid.
2. Agent Direction: Specify the direction the red triangle is facing, which is the orientation of the vertex (pointy corner) of the isosceles triangle. Choose from 'right', 'down', 'left', or 'up'. Clarify that this direction is independent of movement options.
3. Obstacle Position: Identify and report the coordinates of the two obstacles in ascending order. Compare the coordinates by their x-values first. If the x-values are equal, compare by their y-values.
   (a) First Obstacle: Provide the coordinates (x, y) of the first blue ball.
   (b) Second Obstacle: Provide the coordinates (x, y) of the second blue ball.
4. Direction Movable: Evaluate and report whether the agent can move one cell in each specified direction, namely, the neighboring cell in that direction is active and empty (not obstacle or out of bounds):
   • Right (x + 1): Check the cell to the right.
   • Down (y + 1): Check the cell below.
   • Left (x - 1): Check the cell to the left.
   • Up (y - 1): Check the cell above.
   Each direction's feasibility should be reported as 'true' if clear and within the grid, and 'false' otherwise.

Reporting Format: Carefully report each piece of information sequentially, following the format 'name: answer'. Ensure each response is precise and reflects careful verification of the grid details as viewed.

## Boxing

**Prompt:** Here are two consecutive rendered frames from the Atari Boxing environment. The game screen is 160x210 pixels, with (0, 0) at the top-left corner. The x-coordinate increases rightward, and the y-coordinate increases downward. For each frame, estimate the following values as integers:

- The white player's x and y coordinates
- The black player's x and y coordinates

Please carefully determine the following values and give concise answers one by one. Make sure to return an estimated value for each one, even if the task may look challenging.

Follow the reporting format:

- frame 1 white x: estimated_value
- frame 1 white y: estimated_value
- frame 1 black x: estimated_value
- frame 1 black y: estimated_value
- frame 2 white x: estimated_value
- frame 2 white y: estimated_value
- frame 2 black x: estimated_value
- frame 2 black y: estimated_value

## Pong

**Prompt:** Here are four consecutive rendered frames from the Atari Pong environment. The game screen is 84x84 pixels, with (0, 0) at the top-left corner. You need to look at the ball and the paddle in the right. The x-coordinate increases downward, and the y-coordinate increases rightward.

Please determine the following values and give concise answers one by one. Make sure to return an estimated value for each one, even if the task may look challenging.

- x coordinate of the ball relative to the paddle in the first frame
- y coordinate of the ball in the first frame
- x velocity of the ball calculated from the first and second frames
- y velocity of the ball calculated from the first and second frames
- velocity of the paddle calculated from the first and second frames
- acceleration of the paddle calculated from the first three frames
- jerk of the paddle calculated from the four frames

Follow the reporting format in your response.

- x coordinate of the ball: estimated_value
- y coordinate of the ball: estimated_value
- x velocity of the ball: estimated_value
- y velocity of the ball: estimated_value
- velocity of the paddle: estimated_value
- acceleration of the paddle: estimated_value
- jerk of the paddle: estimated_value

## A.4 EXPERIMENTAL DETAILS

**Behavior learning hyperparameters.** For the PPO hyperparameters, we set $4 \cdot 10^6$ total timesteps for PixelCartPole and DoorKey, $10^6$ for DynamicObstacles, $1.5 \cdot 10^7$ for Boxing, and $10^7$ for Pong. For PixelCartPole, DoorKey, and DynamicObstacles, we use 8 vectorized environments, horizon $T = 4096$, 10 epochs for training, batch size of 512, learning rate $3 \cdot 10^{-4}$, entropy coefficient 0.01, and value function coefficient 0.5. For Boxing and Pong, we use 8 vectorized environments, horizon $T = 1024$, 4 epochs for training, batch size of 256, learning rate $3 \cdot 10^{-4}$, entropy coefficient 0.01, and value function coefficient 0.5. For all other hyperparameters, we use the default values from Stable Baselines 3 Raffin et al. (2021).

**Concept learning hyperparameters.** For the concept training, we set 100 epochs with Adam optimizer with the learning rate linearly decaying from $3 \cdot 10^{-4}$ to 0 for each iteration in PixelCartPole, Boxing, and Pong. In DoorKey and DynamicObstacles, we use the same optimizer and initial learning rate, yet set 50 epochs instead and set early stopping with threshold linearly increasing from 10 to 20, to incentivize the concept network not to overfit in earlier iterations. The batch size is 32. We model concept learning for PixelCartPole as a regression problem (minimizing mean squared error). We model concept learning for DoorKey, DynamicObstacles, Boxing,and Pong as classification problems.

**LICORICE-specific hyperparameters.** For LICORICE, we set the ratio for active learning $\tau = 10$, batch size to query labels in the active learning module $b = 20$, and the number of ensemble models $N = 5$ for the first three environments. For the complex environments Boxing and Pong, we choose the ratio for active learning $\tau = 4$, batch size to query labels in the active learning module $b = B_m/5$ and number of ensemble models $N = 5$ to make a balance between performance and speed. The default number of iterations chosen in our algorithm is $M = 4$ for PixelCartPole, $M = 2$ for DoorKey, DynamicObstacles, and $M = 5$ for Boxing and Pong. The sample acceptance rate $p = 0.02$ for PixelCartPoleand $p = 0.1$ for the other four environments. The Random-Q baseline uses the same sample acceptance rate $p$ as LICORICE. In the complex environments Boxing and Pong, we use both the KL-divergence penalty and PPO loss at the end of the algorithm to improve optimization with $\beta = 0.01$, as mentioned in Section 3.

**Computational resources.** We use NVIDIA A6000 and NVIDIA RTX 6000 Ada Generation. Each of our training program uses less than 3GB GPU memory. For Boxingand Pong, each run takes less than 20 hours to finish. For PixelCartPole and DoorKey, each run takes less than 9 hours to finish. For DynamicObstacles, each run takes less than 2 hours to finish.

**Experimental replicability.** For all experiments, we use 5 seeds [123, 456, 789, 1011, 1213] to train the models. We then evaluate on the environment with seed 42 with 100 episodes and take the average of the reward across the episodes and the concept error across the observations within each episode.

| | Algorithm | $R \uparrow$ | $c$ Error $\downarrow$ |
|---|---|---|---|
| PixelCartPole | Sequential-Q | $0.23 \pm 0.09$ | $0.10 \pm 0.04$ |
| | Disagreement-Q | $0.36 \pm 0.20$ | $0.10 \pm 0.04$ |
| | Random-Q | $0.34 \pm 0.14$ | $0.07 \pm 0.02$ |
| | LICORICE | $1.00 \pm 0.00$ | $0.02 \pm 0.00$ |
| | LICORICE+GPT-4o | $0.06 \pm 0.01$ | $0.25 \pm 0.08$ |
| | CPM | $1.00 \pm 0.00$ | $0.01 \pm 0.00$ |
| DoorKey | Sequential-Q | $0.46 \pm 0.09$ | $0.46 \pm 0.05$ |
| | Disagreement-Q | $0.76 \pm 0.12$ | $0.39 \pm 0.07$ |
| | Random-Q | $0.89 \pm 0.03$ | $0.27 \pm 0.08$ |
| | LICORICE | $0.99 \pm 0.01$ | $0.05 \pm 0.01$ |
| | LICORICE+GPT-4o | $0.83 \pm 0.06$ | $0.31 \pm 0.01$ |
| | CPM | $1.00 \pm 0.00$ | $0.00 \pm 0.00$ |
| DynamicObstacles | Sequential-Q | $0.79 \pm 0.44$ | $0.01 \pm 0.01$ |
| | Disagreement-Q | $0.99 \pm 0.01$ | $0.00 \pm 0.00$ |
| | Random-Q | $0.98 \pm 0.01$ | $0.02 \pm 0.01$ |
| | LICORICE | $0.99 \pm 0.00$ | $0.00 \pm 0.00$ |
| | LICORICE+GPT-4o | $0.88 \pm 0.09$ | $0.13 \pm 0.03$ |
| | CPM | $0.97 \pm 0.02$ | $0.00 \pm 0.00$ |
| Boxing | Sequential-Q | $0.87 \pm 0.20$ | $0.32 \pm 0.26$ |
| | Disagreement-Q | $0.83 \pm 0.14$ | $0.20 \pm 0.21$ |
| | Random-Q | $0.96 \pm 0.04$ | $0.14 \pm 0.10$ |
| | LICORICE | $1.00 \pm 0.02$ | $0.05 \pm 0.01$ |
| | LICORICE+GPT-4o | $0.22 \pm 0.01$ | $0.98 \pm 0.01$ |
| | CPM | $1.00 \pm 0.05$ | $0.00 \pm 0.00$ |
| Pong | Sequential-Q | $0.98 \pm 0.02$ | $0.54 \pm 0.03$ |
| | Disagreement-Q | $0.94 \pm 0.09$ | $0.57 \pm 0.05$ |
| | Random-Q | $0.94 \pm 0.05$ | $0.60 \pm 0.04$ |
| | LICORICE | $1.00 \pm 0.02$ | $0.24 \pm 0.16$ |
| | LICORICE+GPT-4o | $0.13 \pm 0.17$ | $0.71 \pm 0.01$ |
| | CPM | $1.00 \pm 0.00$ | $0.00 \pm 0.01$ |

Table 5: Evaluation of the reward $R$ and concept error achieved by all methods in all environments. This is a extended table from Figure 2. The reward is reported as the fraction of the reward upper bound. For PixelCartPole, the $c$ error is the MSE. For the other environments, the $c$ error is 1 - accuracy. The first five algorithms are given a budget of $B = [500, 300, 300, 3000, 5000]$ for each environment, from top to bottom; CPM is given an unlimited budget (in practice, it uses 4M, 4M, 1M, 15M, 10M concept labels respectively). The $\pm$ [value] part shows the standard deviation. For each environment, we bold the highest reward and lowest concept error among the *budget-constrained* methods.

# B    ADDITIONAL RESULTS

In this section, we present both the numerical results as tables from the figures in the main paper, in addition to experimental results not reported in the main paper.

## B.1    BALANCING CONCEPT PERFORMANCE AND ENVIRONMENT REWARD

In Table 5, we present all of the numerical results from Figure 2, including standard deviation. Our method enjoys low variance across all environments in terms of both concept error and reward.

| Environment | $B$ | $M$ | $R \uparrow$ | $c$ Error $\downarrow$ |
|---|---|---|---|---|
| PixelCartPole | 300 | 1 | $0.28 \pm 0.10$ | $0.15 \pm 0.07$ |
| | 400 | 3 | $0.77 \pm 0.22$ | $0.05 \pm 0.01$ |
| | 500 | 4 | $1.00 \pm 0.00$ | $0.02 \pm 0.00$ |
| DoorKey | 100 | 1 | $0.77 \pm 0.04$ | $0.26 \pm 0.04$ |
| | 200 | 2 | $0.93 \pm 0.02$ | $0.09 \pm 0.01$ |
| | 300 | 2 | $0.99 \pm 0.01$ | $0.05 \pm 0.01$ |
| DynamicObstacles | 100 | 1 | $0.96 \pm 0.02$ | $0.04 \pm 0.01$ |
| | 200 | 1 | $0.98 \pm 0.01$ | $0.01 \pm 0.00$ |
| | 300 | 2 | $0.99 \pm 0.00$ | $0.00 \pm 0.00$ |
| Boxing | 1000 | 5 | $0.94 \pm 0.05$ | $0.11 \pm 0.01$ |
| | 2000 | 5 | $0.98 \pm 0.05$ | $0.10 \pm 0.06$ |
| | 3000 | 5 | $1.00 \pm 0.02$ | $0.05 \pm 0.01$ |
| Pong | 1000 | 5 | $0.99 \pm 0.00$ | $0.65 \pm 0.04$ |
| | 3000 | 5 | $1.00 \pm 0.00$ | $0.43 \pm 0.05$ |
| | 5000 | 5 | $1.00 \pm 0.02$ | $0.24 \pm 0.16$ |

Table 6: Performance of LICORICE on all environments for varying budgets. We select $M$ to optimize $R$ - $c$ error without additional weighting since they are observed to have the same magnitude. The reward is reported as the fraction of the reward upper bound. For PixelCartPole, $c$ error is MSE; for the other environments, $c$ error is 1 - accuracy. The $\pm$ [value] part shows the standard deviation. This shows a more complete version of the results in Figure 3.

## B.2 BUDGET ALLOCATION EFFECTIVENESS

In Table 6, we present an extension of the results in Figure 3, including the standard deviation. As expected, as the budget increases, the standard deviation for both the reward and concept error tends to decrease. The one exception is the reward for PixelCartPole. Interestingly, the standard deviation is highest for $B = 400$. We suspect this is because the concept errors may be more critical here, leading to higher variance in the reward performance.

| Environment | $B$ | $M$ | $R \uparrow$ | LICORICE+GPT-4o $c$ Error $\downarrow$ | GPT-4o $c$ Error |
|---|---|---|---|---|---|
| PixelCartPole | 300 | 1 | $0.06 \pm 0.02$ | $0.17 \pm 0.19$ | $0.19 \pm 0.25$ |
| | 400 | 2 | $0.07 \pm 0.02$ | $0.25 \pm 0.15$ | $0.22 \pm 0.15$ |
| | 500 | 2 | $0.06 \pm 0.01$ | $0.25 \pm 0.08$ | $0.24 \pm 0.07$ |
| DoorKey | 100 | 1 | $0.71 \pm 0.04$ | $0.45 \pm 0.04$ | $0.31 \pm 0.03$ |
| | 200 | 2 | $0.74 \pm 0.04$ | $0.33 \pm 0.01$ | $0.31 \pm 0.03$ |
| | 300 | 2 | $0.83 \pm 0.06$ | $0.31 \pm 0.01$ | $0.30 \pm 0.03$ |
| DynamicObstacles | 100 | 1 | $0.27 \pm 0.37$ | $0.20 \pm 0.03$ | $0.16 \pm 0.03$ |
| | 200 | 1 | $0.93 \pm 0.05$ | $0.13 \pm 0.03$ | $0.12 \pm 0.05$ |
| | 300 | 1 | $0.88 \pm 0.09$ | $0.13 \pm 0.03$ | $0.13 \pm 0.03$ |
| Boxing | 1000 | 5 | $0.34 \pm 0.09$ | $0.98 \pm 0.01$ | $0.78 \pm 0.01$ |
| | 2000 | 5 | $0.23 \pm 0.11$ | $0.98 \pm 0.01$ | $0.79 \pm 0.01$ |
| | 3000 | 5 | $0.22 \pm 0.01$ | $0.98 \pm 0.01$ | $0.79 \pm 0.02$ |
| Pong | 1000 | 5 | $0.00 \pm 0.00$ | $0.79 \pm 0.08$ | $0.88 \pm 0.01$ |
| | 3000 | 5 | $0.00 \pm 0.00$ | $0.74 \pm 0.07$ | $0.86 \pm 0.03$ |
| | 5000 | 5 | $0.13 \pm 0.17$ | $0.71 \pm 0.01$ | $0.87 \pm 0.02$ |

Table 7: Performance of LICORICE with GPT-4o integrated into the loop for all environments across different budgets, along with the concept labeling error of GPT-4o as a reference. We select $M$ to optimize $R$ - $c$ error without additional weighting since they are observed to have the same magnitude. GPT-4o $c$ error is evaluated on a random sample of 50 observations from the same rollout set used for LICORICE+GPT-4o, due to API cost constraints. This shows a more complete version of the results in Figure 3.

| Environment | $B$ | Sequential-Q | | Disagreement-Q | | Random-Q | |
|---|---|---|---|---|---|---|---|
| | | $R \uparrow$ | $c$ Error $\downarrow$ | $R \uparrow$ | $c$ Error $\downarrow$ | $R \uparrow$ | $c$ Error $\downarrow$ |
| PixelCartPole | 300 | $0.17 \pm 0.03$ | $0.11 \pm 0.03$ | $0.22 \pm 0.04$ | $0.20 \pm 0.04$ | $0.24 \pm 0.07$ | $0.14 \pm 0.04$ |
| | 400 | $0.23 \pm 0.05$ | $0.10 \pm 0.03$ | $0.26 \pm 0.06$ | $0.13 \pm 0.03$ | $0.28 \pm 0.07$ | $0.09 \pm 0.05$ |
| | 500 | $0.23 \pm 0.07$ | $0.10 \pm 0.03$ | $0.39 \pm 0.28$ | $0.09 \pm 0.03$ | $0.35 \pm 0.16$ | $0.07 \pm 0.02$ |
| DoorKey | 100 | $0.33 \pm 0.07$ | $0.51 \pm 0.08$ | $0.61 \pm 0.14$ | $0.33 \pm 0.07$ | $0.70 \pm 0.07$ | $0.30 \pm 0.02$ |
| | 200 | $0.43 \pm 0.09$ | $0.49 \pm 0.03$ | $0.72 \pm 0.09$ | $0.28 \pm 0.07$ | $0.85 \pm 0.04$ | $0.24 \pm 0.04$ |
| | 300 | $0.51 \pm 0.11$ | $0.45 \pm 0.08$ | $0.76 \pm 0.12$ | $0.41 \pm 0.06$ | $0.90 \pm 0.02$ | $0.27 \pm 0.08$ |
| DynamicObstacles | 100 | $0.43 \pm 0.47$ | $0.04 \pm 0.04$ | $0.82 \pm 0.18$ | $0.04 \pm 0.00$ | $0.34 \pm 0.46$ | $0.05 \pm 0.05$ |
| | 200 | $0.76 \pm 0.43$ | $0.03 \pm 0.03$ | $0.99 \pm 0.02$ | $0.01 \pm 0.00$ | $0.76 \pm 0.42$ | $0.03 \pm 0.02$ |
| | 300 | $0.78 \pm 0.44$ | $0.01 \pm 0.01$ | $0.99 \pm 0.01$ | $0.00 \pm 0.00$ | $0.97 \pm 0.01$ | $0.02 \pm 0.02$ |
| Boxing | 1000 | $0.84 \pm 0.20$ | $0.35 \pm 0.12$ | $0.93 \pm 0.05$ | $0.25 \pm 0.11$ | $0.83 \pm 0.24$ | $0.25 \pm 0.05$ |
| | 2000 | $0.88 \pm 0.12$ | $0.19 \pm 0.11$ | $0.93 \pm 0.06$ | $0.14 \pm 0.04$ | $0.83 \pm 0.11$ | $0.30 \pm 0.18$ |
| | 3000 | $0.87 \pm 0.20$ | $0.32 \pm 0.26$ | $0.83 \pm 0.14$ | $0.20 \pm 0.21$ | $0.96 \pm 0.04$ | $0.14 \pm 0.10$ |
| Pong | 1000 | $0.91 \pm 0.12$ | $0.74 \pm 0.07$ | $0.84 \pm 0.21$ | $0.68 \pm 0.06$ | $0.74 \pm 0.21$ | $0.73 \pm 0.05$ |
| | 3000 | $0.98 \pm 0.02$ | $0.58 \pm 0.08$ | $0.98 \pm 0.01$ | $0.64 \pm 0.03$ | $0.96 \pm 0.04$ | $0.60 \pm 0.04$ |
| | 5000 | $0.98 \pm 0.02$ | $0.54 \pm 0.03$ | $0.94 \pm 0.09$ | $0.57 \pm 0.05$ | $0.94 \pm 0.05$ | $0.60 \pm 0.04$ |

Table 8: Performance of Sequential-Q, Disagreement-Q, and Random-Q for all environments across different budgets. The reward is reported as the fraction of the reward upper bound. For PixelCartPole, $c$ error is MSE; for the other environments, $c$ error is 1 - accuracy. The $\pm$ [value] part shows the standard deviation. This shows a more complete version of the results in Figure 3.

### B.3 Integration with Vision-Language Models

In Table 7, we present an extension of the results in Figure 3 in the main paper, including the standard deviation. Interestingly, the standard deviation for the reward obtained by using LICORICE with GPT-4o as the annotator does not always follow the same trend as shown in Table 6 (when we assume access to a more accurate human annotator). Instead, the standard deviation is relatively consistent for PixelCartPole, regardless of the budget. It steadily decreases for DoorKey, as expected. However, in DynamicObstacles, we see an increase when $B = 300$. We are not sure of the cause of this. Perhaps at this point the algorithm begins overfitting to the errors in the labels from GPT-4o (the

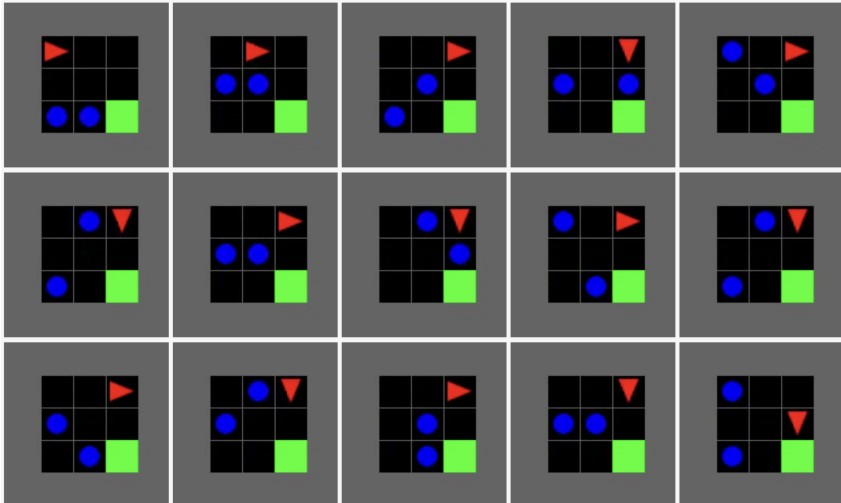

Figure 8: LICORICE+GPT-4o waits until the coast is clear to move to the goal. It appears to make a small mistake in the bottom left, requiring it to wait slightly longer than necessary to navigate to the goal.

concept error rate is the same for 200 and 300 labels). Further investigation is required to understand the underlying factors contributing to this anomaly.

**Different tasks and concepts have various difficulties for GPT-4o.** Table 4 list detailed concept errors for concepts in all environments. In PixelCartPole, cart position and pole angle have smaller errors, while velocities require understanding multiple frames and thus are harder to predict accurately. For DoorKey and DynamicObstacles, different concepts have slightly varying concept errors, indicating visual tasks have different difficulties for GPT-4o. Agent direction has as high as around 0.3 prediction error for both DoorKey and DynamicObstacles. Direction movable is also hard for DoorKey, with a high concept error even if it is a binary concept. We posit it requires the correct understanding of more than one particular object to ensure correctness. The concept accuracies in DoorKey are generally higher than DynamicObstacles, suggesting GPT-4o more struggles with a larger grid.

**A GPT-4o-trained RL agent recovers from a mistake.** In Figure 8, we show an example of a GPT-4o-trained RL agent on DynamicObstacles, in which the agent appears to wait until it is safe to move towards the goal: the green square. The image sequence shows the agent (red triangle) starting from its initial position and moving to the right. It then is cornered by an obstacle (blue circle), then both obstacles. In the bottom left corner frame, it appears to make a mistake by turning to the right, meaning it missed a window to escape. Finally, it moves to the goal when the path is clear in the second-to-last and last frames (bottom right). This behavior highlights that the agent may still learn reasonable behavior even if the concept labels may be incorrect (causing it to make a mistake, as in the bottom left frame).

### B.4 ABLATION

In Table 9, we present an extension of the results in Table 3, including standard deviation. Figure 9 shows all learning curves for ablations in all environments. In PixelCartPole, we clearly see the benefit of iterative strategies on reward: LICORICE, LICORICE-DE and LICORICE-AC consistently increase in reward and converge at high levels, but LICORICE-IT converges at less than $50\%$ of the optimal reward. We also see that LICORICE-IT also achieves higher concept error, indicating that it struggles to learn the larger distribution of concepts induced by a non-optimal policy. In DoorKey, all algorithms steadily increase in reward. However, we can see a dip at around $10^6$ environment steps where we begin the second iteration for LICORICE, LICORICE-AC, and LICORICE-DE. Although LICORICE-IT achieves similar reward to LICORICE, LICORICE achieves lower concept error, which is beneficial for the goal of interpretability. Finally, in DynamicObstacles, LICORICE-

|  | Algorithm | $R \uparrow$ | $c$ Error $\downarrow$ |
|---|---|---|---|
| PixelCartPole | LICORICE-IT | $0.53 \pm 0.26$ | $0.08 \pm 0.03$ |
| | LICORICE-DE | $0.97 \pm 0.05$ | $0.03 \pm 0.01$ |
| | LICORICE-AC | $0.91 \pm 0.10$ | $0.02 \pm 0.01$ |
| | LICORICE | $1.00 \pm 0.00$ | $0.02 \pm 0.00$ |
| DoorKey | LICORICE-IT | $0.99 \pm 0.01$ | $0.09 \pm 0.02$ |
| | LICORICE-DE | $0.78 \pm 0.08$ | $0.20 \pm 0.05$ |
| | LICORICE-AC | $0.82 \pm 0.10$ | $0.16 \pm 0.06$ |
| | LICORICE | $0.99 \pm 0.01$ | $0.05 \pm 0.01$ |
| DynamicObstacles | LICORICE-IT | $1.00 \pm 0.00$ | $0.00 \pm 0.00$ |
| | LICORICE-DE | $0.98 \pm 0.01$ | $0.00 \pm 0.00$ |
| | LICORICE-AC | $0.58 \pm 0.53$ | $0.00 \pm 0.00$ |
| | LICORICE | $0.99 \pm 0.00$ | $0.00 \pm 0.00$ |
| Boxing | LICORICE-IT | $0.98 \pm 0.04$ | $0.14 \pm 0.07$ |
| | LICORICE-DE | $1.00 \pm 0.04$ | $0.04 \pm 0.02$ |
| | LICORICE-AC | $0.99 \pm 0.01$ | $0.09 \pm 0.06$ |
| | LICORICE | $1.00 \pm 0.02$ | $0.05 \pm 0.01$ |

Table 9: Ablation study results for LICORICE in the first four environments, where we study the effect of each component to compare with LICORICE. This shows a more complete version of the results in Table 3.

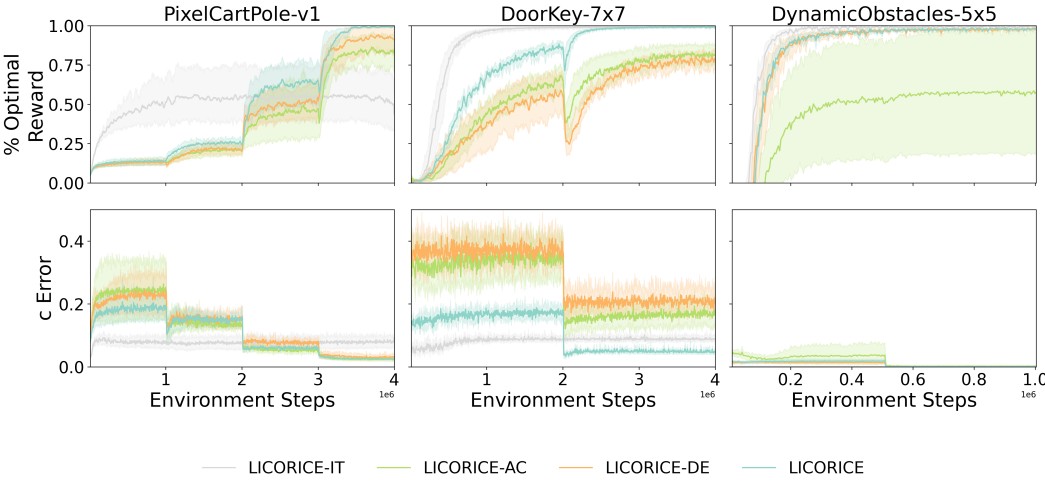

Figure 9: Learning curves for ablations in the first three environments. Shaded region shows 95% CI, calculated using 1000 bootstrap samples.

AC lags behind the most in terms of both reward and concept error. This result indicates that the active learning component is most important for this environment.

**Impact on data diversity.** We hypothesize that data decorrelation enables better performance by increasing the diversity of the training data. As a result, we investigate the role of data decorrelation on state diversity by calculating the average distance to the $k$ nearest samples in the concept label space for all labeled samples. The intuition is that a larger average distance to nearest neighbors indicates samples are more spread out in the space, suggesting higher diversity in the training data. Let $D = \{(s_1, c_1), \ldots, (s_b, c_b)\}$ be the labeled dataset and $\text{Neighbor}(c_i, k)$ be the index set of $k$ nearest neighbors with $k = 10$ for all environments, then we calculate diversity as:

| Environment | LICORICE | | LICORICE-DE | |
| --- | --- | --- | --- | --- |
| | $d$ | $R$ | $d$ | $R$ |
| PixelCartPole | $\mathbf{0.12 \pm 0.00}$ | $1.00 \pm 0.00$ | $0.10 \pm 0.01$ | $0.97 \pm 0.05$ |
| DoorKey | $\mathbf{1.12 \pm 0.18}$ | $0.99 \pm 0.01$ | $0.25 \pm 0.00$ | $0.78 \pm 0.08$ |
| DynamicObstacles | $\mathbf{0.93 \pm 0.04}$ | $0.99 \pm 0.00$ | $\mathbf{0.99 \pm 0.09}$ | $0.98 \pm 0.01$ |
| Boxing | $\mathbf{7.20 \pm 0.20}$ | $0.97 \pm 0.04$ | $6.53 \pm 0.08$ | $0.97 \pm 0.01$ |

Table 10: Training data diversity ($d$) and reward ($R$) with and without data decorrelation. Higher $d$ is better (more diverse). For each environment, the highest $d$ is bolded. When the standard deviations overlap, we bold both.

$$d = \frac{1}{bk} \sum_{i=1}^{b} \sum_{j \in \text{Neighbor}(c_i, k)} ||c_i - c_j||.$$

We present $d$ along with the reward in Table 10 for LICORICE and LICORICE-DE. As hypothesized, decorrelation increases diversity in 3 of the 4 environments. LICORICE achieves higher reward than LICORICE-DE. In DynamicObstacles, the state diversity and reward is similar for both LICORICE and LICORICE-DE. We believe that, in this environment, decorrelation is less critical because neighboring states are inherently more dissimilar due to the dynamic nature of the obstacles.

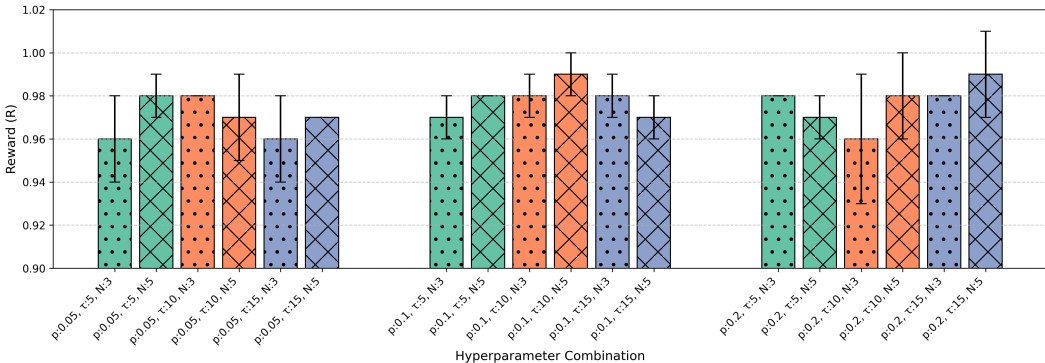

Figure 10: Hyperparameter sensitivity results for DynamicObstacles. Bars represent standard deviation. Here, bars are grouped by the same $p$ values (acceptance probability). Bars of the same color share the same $\tau$ value (budget multiplier), and bars with the same pattern share the same value for $N$ (committee size). Generally, $p = 0.1$ tend to perform better overall with greater stability. Having a larger committee ($N = 5$ vs. $N = 3$) also tends to help with performance.

| Budget | 100 | | 200 | | 300 | |
|---|---|---|---|---|---|---|
| | R | c Error | R | c Error | R | c Error |
| Disagreement | 0.96 | 0.05 | 0.97 | 0.01 | 1.00 | 0.00 |
| Entropy | 0.95 | 0.07 | 0.98 | 0.02 | 0.99 | 0.00 |

Table 11: Comparison of disagreement- and entropy-based active learning strategies for DynamicObstacles under different labeling budgets. There is essentially no difference between the two strategies.

### B.5 HYPERPARAMETER SENSITIVITY

We now seek to understand how sensitive LICORICE is to the choice of hyperparameters. Figure 10 shows the results of this experiment for DynamicObstacles. Interestingly, we find that the main difference between the settings is the variation. Overall, LICORICE is relatively robust to the choice of hyperparameter settings in this environment, with the final average reward ranging between 96% and 99.5% of the optimal. The concept error for all settings was $0.00 \pm 0.00$, so we do not plot it here.

### B.6 CHOICE OF ACQUISITION STRATEGY

We now investigate the impact of the choice of acquisition strategy on both reward and concept accuracy. We implemented an entropy-based disagreement measurement for the committee and studying its effects on a MiniGrid environment. Specifically, we implement the following vote entropy acquisition function: $U(s_i) = -\frac{1}{K} \sum_{k=1}^{K} \sum_{j=1}^{P_k} \frac{V_{ijk}}{N} \log \frac{V_{ijk}}{N}$, where $s_i$ is the input state, $K$ is the number of concepts we learn, $P_k$ is the number of classes for the $k$-th concept, $V_{ijk}$ is the number of votes for the $j$-th class of the $k$-th concept for state $s_i$, and $N$ is the number of committee members. Table 11 shows these results on DynamicObstacles. There is essentially no difference between the two strategies in terms of both reward and concept error.

## C  ADDITIONAL DISCUSSION

### C.1  LIMITATIONS AND FUTURE WORK

While our approach has demonstrated promising results, there are several limitations to be addressed in future work. One significant challenge is the difficulty VLMs face with certain types of concepts, especially continuous variables. This limitation can impact the overall performance of concept-based models, especially in domains where continuous data is prevalent. Furthermore, although VLMs can be successfully used for automatic labeling of some concepts, there are still hallucination issues (Achiam et al., 2023) and other failure cases, such as providing inaccurate counts. We believe that future work that seeks to improve general VLM capabilities and mitigate hallucinations would also help overcome this limitation. Addressing this issue could involve developing specialized techniques or using existing tools and libraries to better complement VLM capabilities.

Another area for future improvement is the refinement of our active learning and sampling schemes. Our current method employs an disagreement-based acquisition function to select the most informative data points for labeling. While this approach is effective, there is potential for exploring more sophisticated active learning strategies, such as incorporating advanced exploration-exploitation trade-offs or leveraging recent advancements in active learning algorithms (Tharwat & Schenck, 2023).

Furthermore, scaling challenges may arise from environmental complexity, such as when only a subset of given concepts are relevant or when learning certain concepts is prerequisite to gathering training data for others. Future research could explore using attention mechanisms (Vaswani et al., 2017) or sparse coding (Olshausen & Field, 1996) to identify relevant concepts. Another exciting direction is the work in human-AI complementarity and learning-to-defer algorithms (Mozannar et al., 2023; Verma et al., 2023) to train an additional classifier for deferring labeling to a person when the chance of an error is high. In our work, we assume a known set of concepts for the environment; future work could investigate the use of human-VLM teams to determine a reasonable concept set for the environment.

Finally, designing a concept-based representation for RL remains an open challenge. Our work provides a few illustrative examples, but the exact design of these representations can significantly impact performance, often for reasons that are not entirely clear — especially when using VLMs as annotators. Prior work (Das et al., 2023) proposed some desiderata for concepts in RL, but future work could refine these principles, especially in the face of VLM annotators. Future work could also include systematically investigating the factors that influence the effectiveness of different concept-based representations in RL. This could involve extensive empirical studies, theoretical analyses, and the development of new design principles that guide the creation of effective concept representations. Understanding these factors better will help in creating more reliable and interpretable RL models, ultimately advancing the field and broadening the applicability of concept-based approaches in various RL tasks.

### C.2  BROADER IMPACTS

**Interpretability in RL.**  Incorporating concept learning with RL presents both positive and negative societal impacts. On the positive side, promoting interpretability and transparency in decision-making fosters trust and accountability. However, in cases where it yields incorrect results, stakeholders might be misled into trusting flawed decisions due to the perceived transparency of the model (Kaur et al., 2020). Unintended misuse could also occur if stakeholders lack the technical expertise to accurately interpret the models, leading to erroneous conclusions and potentially harmful outcomes. To mitigate these risks, an avenue for future work is developing clear guidelines for interpreting these models and tools to scaffold non-experts' understanding of the model outputs.

**Using VLMs for concept labeling.**  On one hand, VLMs have the potential to significantly improve the efficiency and scalability of labeling processes, which can accelerate advancements in various fields. By automating the labeling of large datasets, VLMs can help reduce the time and cost associated with manual labeling. However, there are important ethical and social considerations to address. One major concern is the potential for bias in the concept labels generated by VLMs. If these models are trained on biased or unrepresentative data, they may perpetuate or even amplify existing biases, leading to unfair or discriminatory outcomes. This is particularly problematic in

sensitive applications like hiring, lending, or law enforcement, where biased decisions can have significant negative impacts on individuals and communities. Furthermore, there are privacy concerns related to the data used to train VLMs. Large-scale data collection often involves personal information, and improper handling of this data can lead to privacy violations. To mitigate these risks, future work could include developing robust data governance frameworks to protect individuals' privacy and comply with relevant regulations.

