# OpenReview forum: "LICORICE: Label-Efficient Concept-Based Interpretable Reinforcement Learning"
_ICLR.cc/2025/Conference — ICLR 2025 Poster_

### Official Review · Reviewer_mn27 · 2024-11-01

**Soundness:** 3
**Presentation:** 4
**Contribution:** 3
**Rating:** 6
**Confidence:** 4

**Summary:**

This paper proposes a method of learning a concept-based RL policy that utilizes only a small number of annotations. Prior work introduced the concept-based RL learning for the purpose of obtaining interpretable policies, however, the assumption is that all states are annotated with the concepts. This paper proposes to sample the states and select the most informative of them with a disagreement based active learning strategy. As a result, the proposed method achieves very similar performance with much less annotated data both in terms of policy performance and concept prediction as the original method. Moreover, authors study if VLMs can be used as concept annotators in some environments.

**Strengths:**

- The paper addresses an important problem of making RL policies interpretable for the final users
- The proposed method removes the unrealistic assumption that all the states are annotated with the concepts. The method combines two lines of work: concept-based reinforcement learning with active learning in a novel way
- The paper studies the performance both in the case of using good quality labels and noisy labels (in the experiments it corresponds to using the environment state or VLM).

**Weaknesses:**

I think what I am missing the most in this paper is more explanations on the interpretability of the policy. Given that the novelty of the method is limited in a way that it combines two existing methods, it would be good to demonstrate the importance of the proposed solution more. As the goal is to produce interpretable policies, it would be good to see some examples of interpretable policies, and maybe even verify that users of the policies find them convincing and useful. Also, the paper talks about the "concepts" in a very abstract way and I would like to see some more examples of the concepts to make it more concrete. For example, table 1 shows the number of concepts and their type, but does not communicate that each concept can take more than 100 values in some cases.

Additionally, while the experimental studies are good, I think it could be further improved with some real world experiments (as the original concept-based RL paper presented the experiments in the real world) to increase the impact of the paper.

Minor questions and comments:
- How do you manage to obtain the diverse predictions in the ensemble given that they are trained in the same way?
- More details on the architecture of the policy and concept predictor would be useful
- Data decorrelation seems to be just random subsampling of the data. Is this so or is there any other important aspect? If it is the case, I don't think the authors need to discuss it in so many details.
- Does each state action pair correspond to only one concept? If there are several, how do you decide based on the disagreement in which concept to select the sample to label?

**Questions:**

I would like to understand a bit more about the interpretability of the policies produced by the proposed method.

---

> ### Author Response · Authors · 2024-11-25
> **Official Comment by Authors (1/2)**
>
> We thank the reviewer for acknowledging that our work identifies an important limitation of concept-based models in reinforcement learning and the method is novel. We’ll address your comments below:
>
> > I think what I am missing the most in this paper is more explanations on the interpretability of the policy. Given that the novelty of the method is limited in a way that it combines two existing methods, it would be good to demonstrate the importance of the proposed solution more. As the goal is to produce interpretable policies, it would be good to see some examples of interpretable policies, and maybe even verify that users of the policies find them convincing and useful. Also, the paper talks about the "concepts" in a very abstract way and I would like to see some more examples of the concepts to make it more concrete. For example, table 1 shows the number of concepts and their type, but does not communicate that each concept can take more than 100 values in some cases.
>
> We respectfully disagree that the proposed solution is simply combining two existing methods. Although, at a high level, we leverage tools from the existing literature, these tools do not solve the problem on their own. For example, just using active learning (Disagreement-Q) does not enable us to learn performant RL policies. More generally, CPM assumes unlimited access to an expert, which is not feasible. Understanding and characterizing these challenges is a research contribution on its own. Additionally, our method goes beyond simply applying existing active learning techniques by introducing components to resolve key challenges for interpretable RL training. The results of our ablation study (Table 3) demonstrate that each component is beneficial for achieving high reward and concept accuracy on all four environments.
>
> Regarding interpretability, we appreciate the suggestion to further showcase the interpretability of our approach! To do so, we visualize two examples of how test-time concept intervention can help facilitate proper decision-making. Specifically, we show two examples in DoorKey. We show how the agent incorrectly predicting the door as being closed results in it attempting to open the door with the key (toggle). In contrast, incorrectly predicting the door as being open leads the agent to incorrectly try to move through the doorway. Intervening on this concept enables the agent to correctly choose to open the door or not. We include this visualization in Figure 5 of the updated paper, with the discussion highlighted in blue in Section 4.2, under RQ4.
>
> Good point regarding Table 1. We updated Table 1 to also showcase the number of binary concepts if we binarize the categorical concepts. In our work, concepts are typically properties of the objects present in the environment (e.g., the agent's position and whether the agent is movable along each direction). We are happy to provide additional visualizations in the revised paper. In the meantime, the precise concept definitions can be found in the prompts for the VLM in Appendix A.3.
>
> > Additionally, while the experimental studies are good, I think it could be further improved with some real world experiments (as the original concept-based RL paper presented the experiments in the real world) to increase the impact of the paper.
>
> Thank you for the suggestion! We would like to highlight that the original concept-based RL paper was published at a robotics conference. Although we do not have the ability to run real-world experiments (we would love to be able to!), we have run experiments on another complex Atari environment, Pong, to further investigate LICORICE’s performance. Pong includes 1848 binary concepts in total. Of note: this environment includes flickering effects that cause the paddle to disappear for tens of timesteps after each score increase, increasing the concept prediction difficulty. Using $B=5000$ budgets, the results show LICORICE enjoys a much lower concept error than baselines while achieving almost perfect reward.
>
> |     Method     |   Reward   | Concept Error |
> | :------------: | :--------: | :-----------: |
> |    LICORICE    | 0.99+-0.01 |  0.20+-0.04   |
> |  Sequential-Q  | 0.99+-0.01 |  0.55+-0.04   |
> | Disagreement-Q | 1.00+-0.01 |  0.55+-0.06   |
> |    Random-Q    | 0.95+-0.04 |  0.60+-0.06   |
>
> > How do you manage to obtain the diverse predictions in the ensemble given that they are trained in the same way?
>
> We use different random seeds for initializations and stochasticity of SGD for each model in the ensemble. This difference is sufficient to obtain diverse predictions.
>
> > More details on the architecture of the policy and concept predictor would be useful
>
> Thanks for your interest in the model architecture! Please check the detailed description of the architecture in Appendix A.2.

---

> > ### Author Response · Authors · 2024-11-25
> > **Official Comment by Authors (2/2)**
> >
> > > Data decorrelation seems to be just random subsampling of the data. Is this so or is there any other important aspect? If it is the case, I don't think the authors need to discuss it in so many details.
> >
> > We agree that this technique is simple but effective. We would like to emphasize that states are highly temporally correlated when doing policy rollouts, motivating the decorrelation technique. We find that decorrelation directly impacts state diversity, measured by the average distance to $k$ nearest samples in the concept label space for all labeled samples, which intuitively measures how samples are spread out in the space. In all environments where LICORICE outperforms LICORICE-DE (all but DynamicObstacles), we find that LICORICE’s training data has higher state diversity (Table 1, below). This finding indicates that data decorrelation is indeed important for final performance by improving the diversity of the training data. Details below.
> >
> > Let $D = \lbrace (s_1, c_1), \cdots, (s_b, c_b) \rbrace$ be the labeled dataset and $\text{Neighbor}(c_i,k)$ be the index set of $k$ nearest neighbors, then we calculate diversity as: $$\frac{1}{bk}\sum_{i=1}^b \sum_{j\in \text{Neighbor}(c_i,k)} \Vert c_i-c_j\Vert.$$ We choose $k=10$ for all environments. The results in Table 1 show that, as hypothesized, decorrelation increases diversity in 3 of the 4 environments. This finding aligns with our ablation study: for these same 3 environments, LICORICE achieves higher reward than LICORICE-DE. In DynamicObstacles, the state diversity and reward is similar for both LICORICE and LICORICE-DE. We believe that, in this environment, decorrelation is less critical because neighboring states are inherently more dissimilar due to the dynamic nature of the obstacles.
> >
> > |             | PixelCartPole |  DoorKey   | DynamicObstacles |   Boxing   |
> > | :---------: | :-----------: | :--------: | :--------------: | :--------: |
> > |  LICORICE   |  0.12+-0.00   | 1.12+-0.18 |    0.93+-0.04    | 7.20+-0.20 |
> > | LICORICE-DE |  0.10+-0.01   | 0.25+-0.00 |    0.99+-0.09    | 6.53+-0.08 |
> >
> > Table 1: Training data diversity with and without data decorrelation. Higher is better (more diverse).
> >
> > > Does each state action pair correspond to only one concept? If there are several, how do you decide based on the disagreement in which concept to select the sample to label?
> >
> > For each state, we have several predicted concepts from the ensemble models and calculate the disagreement among them. Then, we choose states with highest disagreement according to the acquisition function to query for concept labels.
> >
> > Once again, thank you for your valuable feedback! We hope we have addressed your comments. Please consider raising your review score if you feel this process has improved the quality of our paper.

---

> > > ### Comment · Reviewer_mn27 · 2024-11-26
> > >
> > > Thank you for providing the answers to my questions and comments. In particular, I appreciate adding more details on interpretability, in my opinion it adds value to demonstrate the benefit of combining two components to address the problem of interpretability. On "data decorrelation", just to note, I did not doubt its necessity, I was more wondering if it is necessary to introduce a new term and explain it in so many details if it is just a random subsampling. Thank you, I don't have any more questions.

---

> > > > ### Author Response · Authors · 2024-11-26
> > > >
> > > > Thanks so much for clarifying!
> > > >
> > > > Regarding interpretability, we agree that the additional details are helpful & appreciate the suggestion. We're happy to add additional examples to the appendix to further strengthen the paper.
> > > >
> > > > Thanks for clarifying the data decorrelation point! We condensed what was previously lines 193-204 to lines 195-204 to make the description of this component more concise.
> > > >
> > > > Thanks again for your review.

---

### Official Review · Reviewer_UqSw · 2024-11-03

**Soundness:** 3
**Presentation:** 3
**Contribution:** 2
**Rating:** 6
**Confidence:** 3

**Summary:**

Current RL methods lack interpretability, which is necessary for stakeholders to comprehend and trust them. Previous work on concept bottlenecks uses a predefined vocabulary, requiring real-time concept annotation when applied to RL. The authors propose a training scheme (LICORICE) that allows RL algorithms to efficiently learn concept-based policies by querying annotators to label only a small set of data.

**Strengths:**

1. The paper is well-written and easy to follow
2. The framework enables interpretability on the state estimation module of the RL framework. Human could interpret the states by reading the intermediate variables.

**Weaknesses:**

1. This method is not general enough. Because environments like cart pole are very simple, it’s possible to use some textual “words” to describe the state. But what about in the real environment with much more states? Also, some of the states are not describable by language.
2. In line 73, the paper mentions relying on human labelers is potentially biasing the model training process. But it seems the paper doesn't mention what is the potential ground truth data (without human labeling) that is less biased. The paper proposes using VLMs (obviously, VLMs are also trained on vision-text data with human knowledge) to automate the concept extraction, but it's unclear that it will not bias the RL model training process as human labelers do?
3. line 76-90, the authors mention two concerns: 1) concept distribution learned from a policy, 2) data diversity.  But in line 91-96, the paper only mentions that the experiments are focused on human annotation and VLM-based annotation. It seems there is not a clear section to address these concerns. Ablation study shows the components are all positive. But it seems no experiment is explicitly talking about training data diversity , e.g., after applying the data decorrelation strategy how does the diversity change?
4. In line 98-99, authors claim they are the first to investigate “limited concept annotation budget for interpretable machine learning”. But there are some papers like "LABEL-FREE CONCEPT BOTTLENECK MODELS" from ICLR2023 propose CBM even without labeled concept data.
5. In those controlled environment (e.g., cartpole), I believe there might be better ways to collect state data (e.g., velocity) rather than query VLM with prompts (Appendix A.3). First, VLMs are not designed to do accurate estimation like pixel loations. Second, using simple heuristics like pattern matching the pixel of the agent may be more efficient and accurate.

**Questions:**

1. The original concept bottleneck models use images as input, which is already easy for human to understand. Also concept bottleneck models are intended to make the final prediction interpretable by decomposing a complex concept (e.g., an object) into its atomic-level properties, then doing classifications on top of them. But the authors mention “”opaque raw inputs”” in line 46-47. Could the authors explain more about this type of input?

---

> ### Author Response · Authors · 2024-11-25
> **Official Comment by Authors (1/n)**
>
> Thank you for taking the time to review our paper! We appreciate the detailed comments on our use of VLMs. We’d like to point out the following additional contributions:
>
> 1. To the best of our knowledge, we are the first to investigate the problem of a limited concept annotation budget for interpretable reinforcement learning. This problem is important because (online) RL is interactive by definition, requiring human-in-the-loop labels during training. As we have shown with the Sequential-Q baseline, we cannot simply deploy a random policy to sample states from the environment and ask a person to label them: this leads to an inability to learn a performant policy.
>
> 2. We introduce LICORICE, a novel training scheme that enables label-efficient learning of concept-based RL policies. LICORICE is designed to address key challenges that arise from the sequential nature of the RL setting. Compared to the state-of-the-art, LICORICE requires 2,000-13,000x fewer queries to achieve the same performance.
>
> 3. We present the first study of VLMs-as-annotators in the context of interpretable reinforcement learning, finding that there is some promise for these methods to help alleviate human annotation burden.
>
> We’ll address your comments below:
>
> > This method is not general enough. Because environments like cart pole are very simple, it’s possible to use some textual “words” to describe the state. But what about in the real environment with much more states? Also, some of the states are not describable by language.
>
> We conduct experiments on four diverse environments, from CartPole to Atari. Although CartPole is seemingly simple, we would like to call your attention to Figure 2, which shows that budget-constrained baselines fail on CartPole, whereas LICORICE does not. The takeaway is that seemingly simple environments may also require a more carefully-designed algorithm to address key challenges, such as human annotation efficiency.
>
> Furthermore, we would like to point out that Atari is the most common benchmark in reinforcement learning, motivating our decision to select Boxing as one of our environments. In fact, Atari is used more in scientific research than the next 8 benchmarks combined [1]. In addition to its popularity, we choose Boxing due to the size of its state space: Boxing has eight ordinal concepts, meaning that the number of states is of the order quintillion (please refer to Table 4 for the concepts and possible values for this environment). If we were to define these concepts as binary variables as in previous work [2], then Boxing would have k=1480 binary concepts. Using the same process for the other two environments with discrete concepts, we would have k=30 binary concepts for DynamicObstacles and k=46 binary concepts for DoorKey. In contrast, the maximum number of concepts studied in the CPM paper is k=28 binary concepts [2], which is fewer than the minimum number of binary concepts studied in our work.
>
> Additionally, we have run experiments on another complex Atari environment, Pong, to further show the generalizability of our method to different complex environments. Pong includes 1848 binary concepts in total. Of note: this environment includes flickering effects that cause the paddle to disappear after each score increase, increasing the concept prediction difficulty. Using $B=5000$ budgets, the results show LICORICE enjoys a much lower concept error than baselines while achieving almost perfect reward.
>
> |     Method     |   Reward   | Concept Error |
> | :------------: | :--------: | :-----------: |
> |    LICORICE    | 0.99+-0.01 |  0.20+-0.04   |
> |  Sequential-Q  | 0.99+-0.01 |  0.55+-0.04   |
> | Disagreement-Q | 1.00+-0.01 |  0.55+-0.06   |
> |    Random-Q    | 0.95+-0.04 |  0.60+-0.06   |
>
> Could you please provide a reference for an environment with states not describable by language?

---

> > ### Author Response · Authors · 2024-11-25
> > **Official Comment by Authors (2/n)**
> >
> > > In line 73, the paper mentions relying on human labelers is potentially biasing the model training process. But it seems the paper doesn't mention what is the potential ground truth data (without human labeling) that is less biased. The paper proposes using VLMs (obviously, VLMs are also trained on vision-text data with human knowledge) to automate the concept extraction, but it's unclear that it will not bias the RL model training process as human labelers do?
> >
> > We would like to clarify our motivation. We rely on human annotations as ground-truth labels to train the concept network. Line 73 serves to motivate the data annotation problem: overwhelming annotation demands can have critical and practical downstream impacts. For example, annotators may become tired and struggle to attend to the task, potentially introducing a larger number of errors in the process.
> >
> > The goal of using VLMs is not because we believe that the errors induced by humans and VLMs are non-overlapping (although this is an interesting area of separate study [3]). Instead, VLMs that perform well on this data annotation task can be used as a substitute for or alongside human annotators, freeing up the human annotator’s time to more carefully validate other parts of the data annotation pipeline or take on other tasks.
> >
> > > line 76-90, the authors mention two concerns: 1) concept distribution learned from a policy, 2) data diversity. But in line 91-96, the paper only mentions that the experiments are focused on human annotation and VLM-based annotation. It seems there is not a clear section to address these concerns. Ablation study shows the components are all positive. But it seems no experiment is explicitly talking about training data diversity , e.g., after applying the data decorrelation strategy how does the diversity change?
> >
> > We are happy to clarify! In line 76-90, we mention three inherent challenges, which LICORICE is designed to address. Specifically, we introduce iterative training to handle the problem of concept learning on off-policy data, data decorrelation to improve the limited training data diversity, and disagreement-based active learning to address the inefficient use of annotation effort. As mentioned in the review, the ablation study proves the effectiveness of all three components.
> >
> > That said, explicitly studying training data diversity is a great suggestion. We investigate the role of data decorrelation on state diversity by calculating the average distance to $k$ nearest samples in the concept label space for all labeled samples. The intuition is that a larger average distance to nearest neighbors indicates samples are more spread out in the space, suggesting higher diversity in the training data. Let $D = \lbrace (s_1, c_1), \cdots, (s_b, c_b) \rbrace$ be the labeled dataset and $\text{Neighbor}(c_i,k)$ be the index set of $k$ nearest neighbors with $k=10$ for all environments, then we calculate diversity as: $$\frac{1}{bk}\sum_{i=1}^b \sum_{j\in \text{Neighbor}(c_i,k)} \Vert c_i-c_j\Vert.$$ The results in Table 1 show that, as hypothesized, decorrelation increases diversity in 3 of the 4 environments. This finding aligns with our ablation study: for these same 3 environments, LICORICE achieves higher reward than LICORICE-DE. In DynamicObstacles, the state diversity and reward is similar for both LICORICE and LICORICE-DE. We believe that, in this environment, decorrelation is less critical because neighboring states are inherently more dissimilar due to the dynamic nature of the obstacles.
> >
> > |             | PixelCartPole |  DoorKey   | DynamicObstacles |   Boxing   |
> > | :---------: | :-----------: | :--------: | :--------------: | :--------: |
> > |  LICORICE   |  0.12+-0.00   | 1.12+-0.18 |    0.93+-0.04    | 7.20+-0.20 |
> > | LICORICE-DE |  0.10+-0.01   | 0.25+-0.00 |    0.99+-0.09    | 6.53+-0.08 |
> >
> > Table 1: Training data diversity with and without data decorrelation. Higher is better (more diverse).
> >
> > > In line 98-99, authors claim they are the first to investigate “limited concept annotation budget for interpretable machine learning”. But there are some papers like "LABEL-FREE CONCEPT BOTTLENECK MODELS" from ICLR2023 propose CBM even without labeled concept data.
> >
> > Thanks for pointing this out! We added a citation and rephrased the claim.

---

> > > ### Author Response · Authors · 2024-11-25
> > > **Official Comment by Authors (3/n)**
> > >
> > > > In those controlled environment (e.g., cartpole), I believe there might be better ways to collect state data (e.g., velocity) rather than query VLM with prompts (Appendix A.3). First, VLMs are not designed to do accurate estimation like pixel loations. Second, using simple heuristics like pattern matching the pixel of the agent may be more efficient and accurate.
> > >
> > > We agree! There exist some environments where using an alternative method to extract concepts could enable more accurate extraction of state data (e.g., CartPole). However, the goal is to explore what happens when one leverages VLMs to perform these annotation tasks. We also want to highlight the main contribution of the work is to propose a training algorithm, LICORICE, to largely decrease the number of human annotations from millions to within 5000. We agree that an exciting future area of study is investigating how to design the pipeline with other vision models or pattern matching.
> > >
> > > > The original concept bottleneck models use images as input, which is already easy for human to understand. Also concept bottleneck models are intended to make the final prediction interpretable by decomposing a complex concept (e.g., an object) into its atomic-level properties, then doing classifications on top of them. But the authors mention “”opaque raw inputs”” in line 46-47. Could the authors explain more about this type of input?
> > >
> > > We are happy to clarify! We indeed use images as input, following the original concept bottleneck model paper. In writing the paper, we also envision real-world environments, like self-driving cars, where the inputs may be multi-modal and challenging to interpret due to their high-dimensionality. We revised these lines to make this more clear.
> > >
> > > Once again, thank you for your valuable feedback! We hope we have addressed your comments. Please consider raising your review score if you feel this process has improved the quality of our paper.
> > >
> > > [1] Delfosse, Quentin, et al. "Interpretable concept bottlenecks to align reinforcement learning agents." arXiv preprint arXiv:2401.05821 (2024).
> > >
> > > [2] Zabounidis, Renos, et al. "Concept learning for interpretable multi-agent reinforcement learning." Conference on Robot Learning. PMLR, 2023.
> > >
> > > [3] Tjuatja, Lindia, et al. "Do llms exhibit human-like response biases? a case study in survey design." Transactions of the Association for Computational Linguistics 12 (2024): 1011-1026.

---

> > > > ### Comment · Reviewer_UqSw · 2024-11-26
> > > >
> > > > Thanks for the very detailed clarification from the authors!
> > > >
> > > > - For the generalization part
> > > >     - The authors argue that some of the virtual environments are complex, and certain baseline methods failed in these scenarios. However, my initial comment was directed at more general and practical RL environments, such as autonomous driving. In those more complex environments, language has its limitation. For instance, challenges arise from high-dimensional sensor data, subtle and transient changes in weather conditions, and complex temporal dynamics (e.g., interactions among multiple moving objects). Addressing these aspects could be a promising direction for future work, as the current algorithm appears to focus primarily on simpler simulated environments.
> > > >
> > > > The authors have addressed most of my concerns and I will update the scores.

---

> > > > > ### Author Response · Authors · 2024-12-02
> > > > >
> > > > > Thank you for your response and clarification of the complex environment setting!
> > > > >
> > > > > >In those more complex environments, language has its limitation. For instance, challenges arise from high-dimensional sensor data, subtle and transient changes in weather conditions, and complex temporal dynamics (e.g., interactions among multiple moving objects).
> > > > >
> > > > > We agree that this would be an interesting direction for future work. Depending on what needs to be interpretable, one way to address this concern is to leverage so-called soft concept models, where concepts that are not describable by language are essentially routed to a non-interpretable channel. However, this design choice can prioritize performance over interpretability and user intervenability.
> > > > >
> > > > > If there is hope of symbolically describing complex interactions, another possibility is to leverage prior work on detecting interactions [1] to analyze the weights connecting concepts to decision-making to discover important higher-order relationships. These discovered interactions can then be used to define new compound concepts that explicitly capture the relationship.
> > > > >
> > > > > >The authors have addressed most of my concerns and I will update the scores.
> > > > >
> > > > > Thank you! Please let us know which comments we can further address. We would be very happy to address them.
> > > > >
> > > > > [1] Tsang, Michael, Dehua Cheng, and Yan Liu. "Detecting statistical interactions from neural network weights." arXiv preprint arXiv:1705.04977 (2017).

---

### Official Review · Reviewer_1gdK · 2024-11-03

**Soundness:** 3
**Presentation:** 3
**Contribution:** 2
**Rating:** 6
**Confidence:** 4

**Summary:**

This paper focuses on concept-based interpretable reinforcement learning. The authors note that existing concept policy model-based methods require a large amount of concept annotations. This is typically too costly in reinforcement learning problems that require extensive interaction for policy learning. To address this issue, the authors introduce active learning, selectively annotating concepts to achieve label-efficient learning.

**Strengths:**

1. This work studies an interesting problem, making RL policies interpretable is important in practical applications.
2. The proposed method indeed addresses the data challenges in the RL policy learning process. This includes separating concept learning from policy learning and using data decorrelation to focus on more diverse states. These aspects make this method more than simply applying traditional (disagreement-based) active learning.
3. The authors validated the effectiveness of the proposed method in several environments.

**Weaknesses:**

1. This work emphasizes interpretable reinforcement learning, but the authors seem to have directly substituted concept learning for interpretability. I understand that concept learning can, to some extent, help improve interpretability, but the authors should demonstrate more "interpretable" results beyond just performance. I noticed that Table 4 provides definitions of concepts for each task, but how do these concepts influence the actions of different policy models? For me, the relationship between concepts and actions is what truly reflects the interpretability of decisions. For example, when the door is locked and the key is to the agent's right, the agent will move to the right; when the door is open, the agent will move towards the door. The authors should provide similar analyses to demonstrate the interpretability of decisions, rather than stopping at the accuracy of concept learning.
2. The current experiments are mainly conducted on limited concepts. Can LICORICE be validated on a larger concept space? I suggest the authors validate their approach on more complex tasks. The chosen active learning baseline is relatively simple, but considering this is a first exploration for label-efficient concept-based reinforcement learning, it is acceptable.
3. Finally, while it may deviate from the topic of interpretable RL, one of the greatest contributions of concept learning to RL is its ability to help RL achieve compositional generalization. I suggest including a related discussion in the section where the concept model is applied to RL.
PDSketch: Integrated domain programming, learning, and planning. NeurIPS'22
Programmatically Grounded, Compositionally Generalizable Robotic Manipulation. ICLR'23

**Questions:**

Regarding the present study, how is the aspect of interpretability substantiated? It would be beneficial to provide the specific methods or analyses employed to demonstrate the interpretable nature of the proposed approach.

----

After the rebuttal, most of my concerns have been addressed. I have raised my score to 6.

---

> ### Author Response · Authors · 2024-11-25
> **Official Comment by Authors (1/2)**
>
> Thank you for your thoughtful review! We’re thrilled that you think that addressing the data challenge in interpretable RL is an interesting and important problem. We’ll address your comments below:
>
> > This work emphasizes interpretable reinforcement learning, but the authors seem to have directly substituted concept learning for interpretability. I understand that concept learning can, to some extent, help improve interpretability, but the authors should demonstrate more "interpretable" results beyond just performance. I noticed that Table 4 provides definitions of concepts for each task, but how do these concepts influence the actions of different policy models? For me, the relationship between concepts and actions is what truly reflects the interpretability of decisions. For example, when the door is locked and the key is to the agent's right, the agent will move to the right; when the door is open, the agent will move towards the door. The authors should provide similar analyses to demonstrate the interpretability of decisions, rather than stopping at the accuracy of concept learning.
>
> Thank you for the suggestion! We completely agree that concept accuracy is not the same as interpretability; however, it is an important condition for bottleneck models to be utilized in practice since erroneous concepts make it harder to interpret the model’s decision.
>
> One benefit of concept bottleneck models is the ability of a person to intervene on the concepts, as originally demonstrated in section 6 of the CBM paper [1]. We show the results of a test-time concept intervention experiment in Figure 4. Specifically, we find that intervening one by one on the concepts of a concept model with additional noise introduced results in increased reward to optimal or near-optimal performance. This experiment also shows that some concepts may not be necessary to learn completely accurately to obtain a high-performing policy (e.g., in CartPole, we can achieve 100% optimal reward by only intervening on 3 of the 4 total concepts).
>
> We appreciate the suggestion to further showcase the interpretability of our approach! To do so, we visualize two examples of how test-time concept intervention can help facilitate proper decision-making. Specifically, we show two examples in DoorKey. We show how the agent incorrectly predicting the door as being closed results in it attempting to open the door with the key (toggle). In contrast, incorrectly predicting the door as being open leads the agent to incorrectly try to move through the doorway. Intervening on this concept enables the agent to correctly choose to open the door or not. We include this visualization in Figure 5 of the updated paper, with the discussion highlighted in blue in Section 4.2, under RQ4.

---

> ### Author Response · Authors · 2024-11-25
> **Official Comment by Authors (2/2)**
>
> > The current experiments are mainly conducted on limited concepts. Can LICORICE be validated on a larger concept space? I suggest the authors validate their approach on more complex tasks. The chosen active learning baseline is relatively simple, but considering this is a first exploration for label-efficient concept-based reinforcement learning, it is acceptable.
>
> Thanks for the comment! First, we want to call attention to the concept space for the two more challenging environments: CartPole and Boxing. In CartPole, the four concepts are continuous-valued, meaning that there is a potentially infinite number of states assuming infinite precision. In Boxing, there are eight ordinal concepts, meaning that the number of possible states is in the order of at least trillions (please refer to Table 4 for the concepts and possible values for this environment). If we were to follow previous work and define these concepts as binary variables, then we would transform each concept value into a separate binary concept, resulting in k=1480 concepts for Boxing. Using the same process for the other two environments, we would have k=30 binary concepts for DynamicObstacles and k=46 binary concepts for DoorKey. In contrast, the maximum number of concepts studied in the CPM paper is k=28 binary concepts [2], which is fewer than the minimum number of binary concepts studied in our work.
>
> Second, we have run experiments on another complex Atari environment, Pong, to further show LICORICE’s performance. The concepts for Pong are borrowed from the VIPER paper [3], including position $(x, y)$ and velocity $(v_x, v_y)$ of the ball, and the position $y_p$, velocity $v_p$, acceleration $a_p$, and jerk $j_p$ of the player's paddle (in total 1848 binary concepts). Of note: this environment includes flickering effects that cause the paddle to disappear after each score increase, increasing the concept prediction difficulty. Using $B=5000$ budgets, the results show LICORICE enjoys a much lower concept error than baselines while achieving almost perfect reward.
>
> |     Method     |   Reward   | Concept Error |
> | :------------: | :--------: | :-----------: |
> |    LICORICE    | 0.99+-0.01 |  0.20+-0.04   |
> |  Sequential-Q  | 0.99+-0.01 |  0.55+-0.04   |
> | Disagreement-Q | 1.00+-0.01 |  0.55+-0.06   |
> |    Random-Q    | 0.95+-0.04 |  0.60+-0.06   |
>
> > Finally, while it may deviate from the topic of interpretable RL, one of the greatest contributions of concept learning to RL is its ability to help RL achieve compositional generalization. I suggest including a related discussion in the section where the concept model is applied to RL.
>
> Thank you for the suggestion! We added these papers to the updated draft and updated the related work section accordingly. Compositional generalization in the context of concepts and, more generally, concept representation learning for RL is a very interesting direction to explore in future work.
>
> Once again, thank you for your valuable feedback! We hope we have addressed your comments. Please consider raising your review score if you feel this process has improved the quality of our paper.
>
> [1] Koh, Pang Wei, et al. "Concept bottleneck models." International conference on machine learning. PMLR, 2020.
>
> [2] Zabounidis, Renos, et al. "Concept learning for interpretable multi-agent reinforcement learning." Conference on Robot Learning. PMLR, 2023.
>
> [3] Bastani, Osbert, Yewen Pu, and Armando Solar-Lezama. "Verifiable reinforcement learning via policy extraction." Advances in neural information processing systems 31 (2018).

---

> ### Comment · Reviewer_1gdK · 2024-11-29
>
> Thanks for your response and additional results. Please see my specific response below.
>
> **Concept learning and interpretability**
> If I understand correctly, test-time concept intervention operates during the testing phase. Due to the proposed method, or rather the concept model's ability to provide concept-based annotations for human understanding, humans/VLMs/LLMs can intervene to help improve decision-making performance. This is reasonable and demonstrates the advantages of interpretable models.
>
> However, this does not directly address my question about how well or poorly the method performs in terms of interpretability—similar to how we evaluate a method's performance using metrics. I have further read some interpretable RL methods proposed in recent years, and it seems there isn't yet a good evaluation method for interpretability. Does the author have any thoughts on how to evaluate the interpretability of decisions beyond concept learning performance, or perhaps I haven't properly understood the role of test-time concept intervention in this aspect?
>
>
>
> **Validation on a larger concept space [solved]**
>
> **​Literatures**
> A recently released paper [1] follows previous research on RL with concept learning (PDSketch, NeurIPS'22). Although it would be unreasonable to require method comparisons now, considering that it also focuses on the data efficiency problem in concept learning, I suggest providing a discussion in the literature part.
> [1] Learning for Long-Horizon Planning via Neuro-Symbolic Abductive Imitation. arXiv preprint arXiv:2411.18201, 2024.
>
> Overall, the author's detailed responses have addressed most of my concerns. Considering other reviewers' positive reception of this work, I do not oppose the paper's acceptance.

---

> ### Author Response · Authors · 2024-12-01
>
> Yes, this is exactly right. Test-time concept interventions enable users to not only improve the performance but also interrogate the policy. For example, a user could ask the counterfactual question: what would happen in this state if the door was locked? By intervening on the door locked concept, the user can directly understand and interpret the impact of this change on the policy.
>
> More broadly, we agree that assessing interpretability is challenging. We could utilize proxies like how rapidly the model improves due to test-time intervention to partly indicate interpretability, yet the most fundamental evaluation should require humans' reception in the context. Given the novelty of concept-based explanations for RL, we believe that conducting a large-scale user study to investigate whether people can accurately predict the outcomes of concept changes, policy behavior, and more would be a promising future direction for assessing interpretability in this context.
>
> Thanks for the reference! One key difference is this work studies sample efficiency in terms of human task demonstrations, whereas we study sample efficiency in terms of human concept labels. However, this paper is very interesting, and we are happy to include it in the related work section.

---

> > ### Comment · Reviewer_1gdK · 2024-12-02
> >
> > Thank you for the response. I have updated the score.

---

> > > ### Author Response · Authors · 2024-12-02
> > >
> > > Thank you again for your thorough review.
> > >
> > > If you have unresolved questions, we would be happy to address them.

---

### Official Review · Reviewer_AVBF · 2024-11-04

**Soundness:** 4
**Presentation:** 4
**Contribution:** 2
**Rating:** 8
**Confidence:** 4

**Summary:**

This papers proposes an active learning scheme to minimize the required concept labels for training concept policy models, which are RL policy models with concept bottlenecks. The method is simple and requires training an ensemble of concept models to estimate the disagreement for prioritizing samples for labeling. Extensive experiments show the label-efficiency of their proposed scheme over the non-active learning golden CPM, and other simple label-restricted baselines. The authors also explored replacing the concept labeling process with a VLM (gpt-4o) and find mixed results with its success.

**Strengths:**

## Originality

All the subcomponents of the proposed method are well-established techniques from their respective fields but chaining them together is an original contribution. The problem of label-efficiency has not been studied for CPM as well (even under-explored in general CBMs).

## Quality

The paper is extremely well written. Common questions one might have are all answered and extensively studied. This includes basic ablation studies of each component of the technique, to all the possible things one might do to a model with concept bottlenecks (e.g. intervention, automate concept labeling). The detailed documentation of experiment design (particularly the specification of how the validation set is curated in the algorithm) reflects the rigor of the work. Well done.

## Clarity

The problem statement is well formulated. The motivation is somewhat justified (though I do highly doubt whether label efficiency is the main problem for adopting CPM in place of regular policy networks in RL settings. I digress). The algorithm is well documented. Assumptions and settings are clearly stated.

## Significance

The weakest part of this paper might be its significance. It is true that interpretability via a concept bottleneck would benefit human-understanding of the decision making of RL agents. However, it does not seem like data scarcity is the biggest issue for implementing CPMs, as current settings with concepts defined all come with sufficient concept labels. If there actually exists a real life scenario where the concept labels are hard to obtain, perhaps the active learning setting makes more sense. That being said, the significant performance improvement of LICORICE over the random baseline is non-negligible.

**Weaknesses:**

Please refer to the questions section.

**Questions:**

Here are a couple of clarification questions
* In P4L120, Line18 of the algorithm states "Continue training the concept network g on Dtrain, using Dval for early stopping". Does Dtrain refer to only the new data added in this iteration, or the entire dataset collected thus far?
* In P6L285, the authors state "In Disagreement-Q, the agent similarly spends its budget at the beginning of its learning process". What does the beginning of its learning process refer exactly?
* In P9L449, Figure 3 only shows the performance of LICORICE and its variants. Including the 3 naive baselines (particularly random) would help readers understand how LICORICE relatively scale with respect to budget.

---

> ### Author Response · Authors · 2024-11-25
>
> We appreciate your detailed and constructive review! Regarding significance, we agree that current benchmarks enable easy extraction of concepts by effectively opening the black box of the simulator. However, real-world applications like robotics or multi-agent settings with emergent concepts like high-level strategies would require human annotation. More broadly, as you have noted, the problem of label efficiency is underexplored for concept models in general. By focusing on the neglected area of data efficiency in interpretable and interactive systems, we hope that this work can open up a new avenue of research on human-centered challenges in interpretable machine learning. We would now like to take this opportunity to clarify some of the questions you raise.
>
> > In P4L120, Line18 of the algorithm states "Continue training the concept network g on Dtrain, using Dval for early stopping". Does Dtrain refer to only the new data added in this iteration, or the entire dataset collected thus far?
>
> Dtrain refers to the entire dataset collected, since we perform data aggregation (please see L17 of the pseudocode).
>
> > In P6L285, the authors state "In Disagreement-Q, the agent similarly spends its budget at the beginning of its learning process". What does the beginning of its learning process refer exactly?
>
> Thanks for the question! The Disagreement-Q agent spends all the budget on the *initial policy rollout*, with an additional active learning function $\alpha(\cdot)$ to choose samples. In contrast, Sequential-Q does not include this active learning function, so it queries the first B states encountered.
>
> > In P9L449, Figure 3 only shows the performance of LICORICE and its variants. Including the 3 naive baselines (particularly random) would help readers understand how LICORICE relatively scale with respect to budget.
>
> Great suggestion. We revised Figure 3 to additionally include the 3 baselines. Across all environments, LICORICE is the most query efficient, overall achieving higher reward and lower concept error than the baselines across all budget levels. This study reveals another interesting finding: LICORICE and Random-Q achieve similar concept error for DynamicObstacles for all budgets. However, LICORICE achieves higher reward for lower budgets (100, 200) and comparable reward when the budget is 300. This finding suggests that LICORICE can learn more efficiently through strategic query selection compared to random sampling.
>
> Once again, thank you for your valuable feedback! We hope we have addressed your comments.

---

> > ### Comment · Reviewer_AVBF · 2024-11-26
> >
> > Thank you for the response. I have no further questions.

---

### Official Review · Reviewer_DRKb · 2024-11-05

**Soundness:** 3
**Presentation:** 3
**Contribution:** 3
**Rating:** 6
**Confidence:** 3

**Summary:**

The paper introduces LICORICE, a training paradigm for efficient concept-based reinforcement learning. It reduces the need for extensive annotations. LICORICE uses active learning to label only informative data and applies data decorrelation for diverse samples. Experiment results show that LICORICE maintains performance while reducing annotation requirements.

**Strengths:**

1. The paper is generally well-written and structured, followed by the motivation and explanation.
2. By combining iterative training, data decorrelation, and disagreement-based active learning, LICORICE significantly reduces annotation requirements without sacrificing performance.
3. The model achieves better results compared to baseline methods.

**Weaknesses:**

1. During concept learning, $g$ is trained using $D_{train}$. However, as the agent's behavior changes, it may encounter new concepts during behavior learning. This can lead to a distribution mismatch between the data used for concept learning and the actual states visited during behavior learning.
2. The authors mention that LICORICE reduces the need for extensive annotations. I agree with this point. However, I think right now people care more about the computation efficiency instead of the cost of labeling data. It would be better to show the training efficiency with less budget.
3. In the budget reduction session, the authors initially use a large budget and then reduce it to a smaller one. I suggest starting with a small budget and gradually increasing it to identify the minimum human labeling effort required to maintain an acceptable reward.

**Questions:**

1. In Algorithm 1, please clarify the definitions of $s$ and $c$.
2. In line 187, there is an extra “the.”

**Details Of Ethics Concerns:**

No ethic concern

---

> ### Author Response · Authors · 2024-11-25
>
> Thank you for your thoughtful review! We’ll address your comments below:
>
> > During concept learning, g  is trained using D_{train}. However, as the agent's behavior changes, it may encounter new concepts during behavior learning. This can lead to a distribution mismatch between the data used for concept learning and the actual states visited during behavior learning.
>
> We agree! We mention this challenge in Section 3, starting from line 153. To alleviate the issue of concept distribution shift, we propose iterative training so that new collected concepts are on-policy and helpful for concept training. In our ablation study (Section 4, starting from line 460), we find that iterative training primarily leads to higher reward in the two environments requiring more precise control. We also find that it leads to reduced concept error for three of the four tested environments, indicating that it can indeed help with the challenge of concept distribution shift.
>
> > The authors mention that LICORICE reduces the need for extensive annotations. I agree with this point. However, I think right now people care more about the computation efficiency instead of the cost of labeling data. It would be better to show the training efficiency with less budget.
>
> Thank you for this thoughtful point. We want to clarify that computational efficiency was carefully considered in our work. As shown in Figure 9, all methods (including LICORICE and baselines) use the same number of environment timesteps and have comparable computational costs. All experiments across the 4 environments were completed within 12 hours on a single A6000 GPU, which demonstrates LICORICE's practical efficiency for real-world deployment.
>
> > In the budget reduction session, the authors initially use a large budget and then reduce it to a smaller one. I suggest starting with a small budget and gradually increasing it to identify the minimum human labeling effort required to maintain an acceptable reward.
>
> Thanks for the suggestion. We have revised section 4, RQ 3 accordingly. Changes are marked in blue and the results for this experiment are in Figure 3. We define acceptable to mean 99% of the optimal reward. To study how efficiently the various algorithms can leverage limited labeling budgets, we have also included the baselines in this figure. We find that LICORICE is at least as label-efficient than the baselines, typically yielding much higher performance for the same label budget.
>
> > In Algorithm 1, please clarify the definitions of s and c. In line 187, there is an extra “the.”
>
> In Algorithm 1, s means the state and c refers to the queried concept label. We removed the additional “the” in the revised paper.
>
> Once again, thank you for your valuable feedback! We hope we have addressed your concerns. Please consider raising your review score if you feel this process has improved the quality of our paper.

---

> > ### Comment · Reviewer_DRKb · 2024-11-27
> >
> > Thanks for the rebuttal. I have no further questions.

---

> > > ### Author Response · Authors · 2024-12-02
> > >
> > > Thank you again for your thoughtful review and feedback. If you have any unresolved questions, we would be happy to address them.

---

### Author Response · Authors · 2024-11-25
**Global response**

We thank the reviewers for their time and effort in reviewing our paper. We are pleased that our work is considered well-written and easy to follow (DRKb, AVBF, UqSw, mn27), novel in addressing the important challenge of data efficiency in concept-based RL (AVBF, mn27), effective at significantly reducing annotation requirements while maintaining performance (DRKb, AVBF), and sound in its technical approach with thorough empirical validation (DRKb, AVBF).

We have made several changes to the paper. We have also provided a revised draft, with changes noted in blue.

1. Regarding interpretability (1gdK, mn27), we have moved the test-time concept intervention experiments to the main paper (previously in appendix) and provided concrete examples showing how concept predictions influence agent decisions (Figure 5).

2. Regarding environment complexity (UqSw, 1gdK), we note that our environments have large state spaces: Boxing has 1.27 quintillion possible states. Converting to binary concepts (as in prior work) would yield 1,480 concepts for Boxing, greatly exceeding the scale of prior work (e.g., max 28 binary concepts in CPM).

3. To further demonstrate scalability (UqSw, 1gdK), we provide preliminary results on the Pong environment. Using $B=5000$ budgets, the results show LICORICE enjoys a much lower concept error than baselines while achieving almost perfect reward. We will include these results in the next updated version.

4. Regarding the budget study, we have updated it to start from a small budget (DRKb) and added the baselines (AVBF). We find that, overall, LICORICE more efficiently leverages limited queries to achieve higher reward and lower concept error compared with the baselines.

---

### Meta-Review · Area_Chair_es64 · 2024-12-20

**Metareview:**

The paper is well-written and addresses the important problem of interpretable RL. It is supported by good empirical results in several environments. The paper received unanimously positive reviews with ratings of {6, 6, 8, 6, 6}, and the authors effectively addressed the reviewers' questions during the rebuttal. Therefore, this paper is recommended for acceptance.

**Additional Comments On Reviewer Discussion:**

All reviewers participated in discussions with the authors during the rebuttal. The authors addressed the reviewers' questions, leading two reviewers to update their scores.

---

### Decision · Program_Chairs · 2025-01-22

Accept (Poster)